# Two distinct waves of transcriptome and translatome changes drive *Drosophila* germline stem cell differentiation

Tamsin J Samuels [ID][1,2,3], Jinghua Gui[1,3], Daniel Gebert[1,2] & Felipe Karam Teixeira [ID][1,2 ✉]

## Abstract

The tight control of fate transitions during stem cell differentiation is essential for proper tissue development and maintenance. However, the challenges in studying sparsely distributed adult stem cells in a systematic manner have hindered efforts to identify how the multilayered regulation of gene expression programs orchestrates stem cell differentiation in vivo. Here, we synchronised *Drosophila* female germline stem cell (GSC) differentiation in vivo to perform in-depth transcriptome and translatome analyses at high temporal resolution. This characterisation revealed widespread and dynamic changes in mRNA level, promoter usage, exon inclusion, and translation efficiency. Transient expression of the master regulator, Bam, drives a first wave of expression changes, primarily modifying the cell cycle program. Surprisingly, as Bam levels recede, differentiating cells return to a remarkably stem cell-like transcription and translation program, with a few crucial changes feeding into a second phase driving terminal differentiation to form the oocyte. Altogether, these findings reveal that rather than a unidirectional accumulation of changes, the in vivo differentiation of stem cells relies on distinctly regulated and developmentally sequential waves.

**Keywords** Differentiation; Transcription; Translation; Germline; Drosophila
**Subject Categories** Chromatin, Transcription & Genomics; Development; Translation & Protein Quality

## Introduction

Adult stem cells divide repeatedly, replenishing the stem cell population while producing daughter cells that undergo a change in fate and differentiate to specialised cell types. The initial fate change is often driven by a master differentiation factor, which may be either asymmetrically segregated to one daughter cell, or upregulated in response to exclusion from a stem cell niche (Morrison and Spradling, 2008; Homem and Knoblich, 2012). Many master differentiation factors have been identified by their mutant phenotypes: blocking differentiation while accumulating stem cells. However, understanding the changes downstream of the master regulator is much more complex and experimentally challenging, especially in adult stem cells such as in the hematopoietic system, gut and skin (Watt, 2001; Micchelli and Perrimon, 2006; Blanpain and Fuchs, 2006; Comazzetto et al, 2021). Adult stem cells are found in small numbers and can be quiescent for long periods, making it difficult to purify different stages of differentiation from living organisms, or to culture them in vitro.

*Drosophila* ovarian germline stem cells (GSCs) divide throughout adulthood to produce the oocytes, and are one of the best-characterised in vivo stem cell systems (Spradling et al, 2011). GSCs are located at the anterior of the ovary in a structure called the germarium, which houses the entire differentiation process (Fig. 1A) (Xie and Spradling, 2000). GSCs are maintained in a stem cell niche, receiving Dpp/BMP signalling from the somatic cap cells. With each division, one cell is retained in the stem cell niche, while the other is normally excluded from the niche and initiates differentiation. The differentiating daughter cell (cystoblast, CB) undergoes four mitotic divisions with incomplete cytokinesis, resulting in a 16-cell cyst (16cc) interconnected by a cytoplasmic bridge and a structure called the fusome. One of the 16 cells is selected to be the oocyte and meiosis is initiated, while the other 15 cells become nurse cells. Together with surrounding somatic epithelial cells, the 16cc will then form an egg chamber, beginning the process of vitellogenesis to produce the egg.

The master regulator of differentiation in GSCs is *bag-of-marbles* (*bam*) (McKearin and Spradling, 1990). Dpp signalling from the somatic niche represses *bam* transcription in GSCs, so when a daughter cell is excluded from the niche, *bam* is transcriptionally upregulated and drives initiation of differentiation (Chen and McKearin, 2003b, 2003a). *bam* mutants cannot differentiate, and so mutant ovaries accumulate GSC-like cells, the phenotype for which the gene is named. After Bam upregulation initiates differentiation, an accumulation of cellular changes is observed during differentiation, leading the single-nucleated GSC to develop into the complex and structured 16cc syncytium. The first obvious change is a switch from complete cytokinesis in the GSC, which depends on ribosome biogenesis

---

[1]Department of Genetics, University of Cambridge, Downing Street, CB2 3EH Cambridge, UK. [2]Department of Physiology, Development and Neuroscience, University of Cambridge, Downing Street, CB2 3DY Cambridge, UK. [3]These authors contributed equally: Tamsin J Samuels, Jinghua Gui. ✉E-mail: fk319@cam.ac.uk

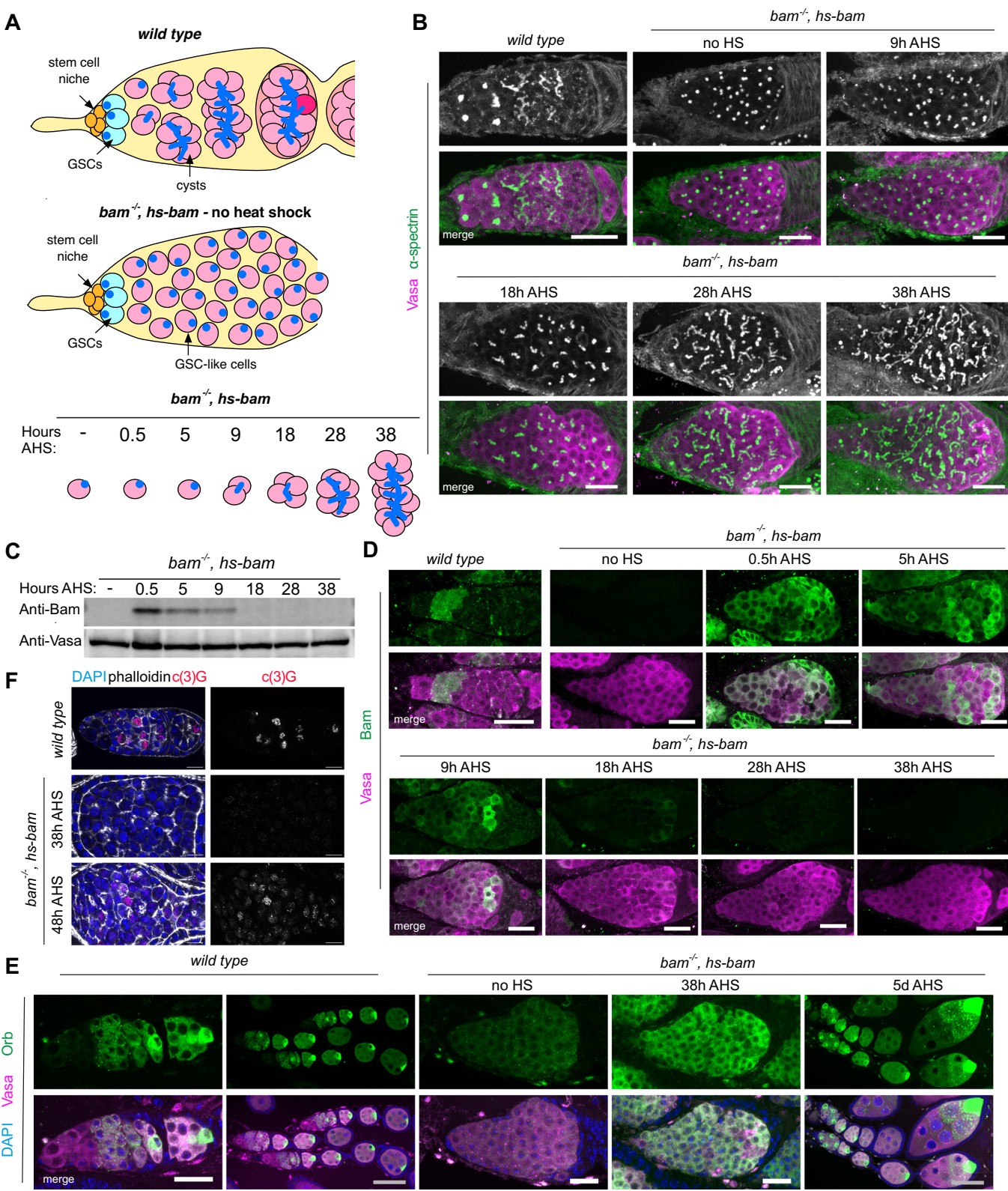

(Sanchez et al, 2016), to incomplete cytokinesis in differentiating cysts via decay of the ESCRT-II machinery (Mathieu et al, 2013; Matias et al, 2015; Eikenes et al, 2015; Mathieu et al, 2022). Ribosome biogenesis is high in GSCs, becoming increasingly suppressed during differentiation (Neumüller et al, 2008; Zhang et al, 2014). At the same time, we have reported that global translation levels are upregulated in early differentiating cells compared to GSCs (Sanchez et al, 2016) and this is necessary for

◄ **Figure 1. Establishing a robust protocol to synchronise GSC differentiation in vivo.**

(**A**) Sketch illustrating *wild type* differentiation (top), and the accumulation of GSC-like cells in the *bam, hs-bam* ovaries used in the synchronisation experiment (middle). Heat shock induces Bam expression resulting in entry into differentiation, such that samples collected at the given times after heat shock (AHS) are enriched for particular stages of differentiation (bottom). (**B**) For example, germaria from *wild type* or *bam, hs-bam* ovaries at the specified times AHS. Stained with immunofluorescence (IF) against α-spectrin (fusome, green and grayscale) and Vasa (germline, magenta). Images are a maximum z projection of 20 slices at 0.5 μm distance. (**C**) Western blot using lysate from *bam, hs-bam* ovaries at specified times AHS. Probed for Bam (top) and Vasa (bottom). (**D**) IF against Bam (green) and Vasa (magenta) in germaria at given time points AHS. (**E**) IF against Orb (green) as a marker for terminal differentiation. DAPI (DNA, blue) and Vasa (magenta) are also stained. (**F**) IF against C(3)G (red) as a marker of the synaptonemal complex. DAPI (DNA, blue) is also stained. Scale bars in (**B, D, E**) (white) are 20 μm, in (**E**) (grey) are 50 μm, in (**F**) are 10 μm. Source data are available online for this figure.

differentiation (Gui et al, 2023). During the syncytial divisions, the microtubule network reorganises such that the minus ends orient to the fusome and then the future oocyte (Grieder et al, 2000), and this will be used to transport the *oskar* (*osk*) mRNA (Ephrussi et al, 1991), the oo18 RNA-binding protein (Orb) (Lantz et al, 1994) and mitochondria (Cox and Spradling, 2003) to the oocyte. The mitochondria also undergo morphological changes, increasing the number of cristae during differentiation through the dimerisation of ATP synthase (Teixeira et al, 2015). Homologous chromosomes are unpaired in GSCs and undergo pairing from the 2cc to 8cc stages such that at the end of differentiation at the 16cc stage, chromosomes are paired in time for the initiation of meiosis (Christophorou et al, 2013). These examples illustrate the extent and complexity of cellular changes during GSC differentiation but how the regulatory processes direct and orchestrate them at a gene expression level remains unknown.

The available methodology has limited the study of gene expression changes and regulation during GSC differentiation, as sequencing experiments on whole ovaries do not capture the early stages of differentiation. Systematic RNA imaging has been used genome-wide to study gene expression patterns in the ovary (Jambor et al, 2015), but these scalable methods have not achieved sufficient resolution in the germarium, where stem cell differentiation occurs. Developmental staging, genetic tools and FACS purification have been used to study the transcriptome of GSCs (Kai et al, 2005; DeLuca et al, 2020), which have been compared to mixed differentiating cysts (Cash and Andrews, 2012; Blatt et al, 2021) and early follicle stages (Pang et al, 2023). Most recently, single-cell RNA sequencing has been performed on whole ovaries by several groups (Jevitt et al, 2020; Rust et al, 2020; Slaidina et al, 2021; Sun et al, 2023), identifying mRNA expression signatures of undifferentiated germline cells, immature and mature nurse cells and oocytes, as well as many different somatic cell types of the ovary. Pseudotime analysis has been used to subcluster germline cells but, due to the low depth of existing single-cell technologies, present a superficial view of the changes in the transcriptome. Furthermore, the technologies to assess translation regulation at a single-cell level are still in their infancy (VanInsberghe et al, 2021). Therefore, the genome-wide assessment of the transcriptome and translatome with high temporal resolution throughout differentiation of any in vivo stem cell system has not been achieved, hindering our understanding of how complex changes are coordinated downstream of master regulators.

We have leveraged established genetic tools to synchronise GSC differentiation in vivo (Ohlstein and McKearin, 1997; Kim et al, 2017; Lu et al, 2020). Our protocol allows the collection of sufficient material for genome-wide experiments, including RNA-seq (measuring the transcriptome) and Ribo-seq (measuring the translatome) at high temporal resolution during the GSC differentiation process (6 time points from GSC to 16cc). During the differentiation process from stem cell to 16cc, we find that the transcriptome undergoes a more significant transformation than anticipated, but nevertheless, the translatome undergoes 3-fold more changes than the transcriptome. Our data facilitate insights into the mechanisms behind this regulation during differentiation, including changes in promoter usage, splicing and translation efficiency, and will be a valuable resource for developmental biologists. Surprisingly, rather than a unidirectional accumulation of changes throughout differentiation, our data reveal two waves of gene expression changes at the level of both the transcriptome and translatome. As the master differentiation factor, Bam, resolves to levels comparable to the stem cell, the differentiating cell returns to a remarkably stem cell-like program of gene expression, with the differential regulation of just a small number of key genes remaining. We suggest that these crucial differences determine the developmental trajectory to terminal differentiation via the second wave of expression changes.

## Results

### Establishing a robust protocol to synchronise GSC differentiation in vivo

Female *bam*$^{-/-}$ mutant flies accumulate GSC-like cells that fill their ovaries, and these samples have been used previously to examine the transcriptome and translatome of the GSC (Kai et al, 2005; Wilcockson and Ashe, 2019; McCarthy et al, 2022). To measure gene expression changes with high developmental resolution during differentiation, we took inspiration from methods that induce germline differentiation by coupling a *bam*$^{-/-}$ mutant with a transgene that can drive differentiation through brief restoration of Bam via a heat shock promoter (*bam*$^{-/-}$,*hs-bam*) (Kai and Spradling, 2004; Kim et al, 2017; Lu et al, 2020; Blatt et al, 2021). Previous studies in the adult ovary have found that the introduction of Bam via a heat shock promoter can drive differentiation in *bam*$^{-/-}$ mutants such that ovaries contain well-formed egg chambers after 6–8 days (Ohlstein and McKearin, 1997). Therefore we asked whether shorter time points after Bam induction in a *bam*$^{-/-}$,*hs-bam* ovary could be used to collect samples enriched for different stages of differentiation (Fig. 1A, "Methods"). We optimised the heat shock protocol to maximise the number of GSC-like cells entering differentiation, while minimising fly death due to heat stress.

Confocal microscopy was used to assign time points representing each stage from undifferentiated GSC to 16cc (terminal

differentiation). We used the branching of the fusome (stained for alpha-spectrin) (Lin et al, 1994; Cuevas and Spradling, 1998) to observe the progression of differentiation at each time point after heat shock (AHS) (Fig. 1B). Based on this branching, we assigned each time point to an approximate stage of differentiation: "no HS" (*bam* mutant), 5 h (cystoblast, CB), 9 h (2cc), 18 h (4cc), 28 h (8cc), and 38 h (16cc) AHS (Fig. 1A). This time course provides a framework for studying GSC differentiation, but it is important to note that we observed heterogeneity in cyst stages, which increased at later time points.

In *wild type* ovaries, Bam protein is not expressed in the GSC, but is upregulated during early differentiation (McKearin and Ohlstein, 1995). To visualise the kinetics of Bam expression induced by our heat shock protocol, we performed western blot and immunofluorescence (IF) analyses across our time course. Performing western blots on whole ovaries revealed that Bam protein was strongly induced at 0.5 h AHS and then declined to levels similar to those observed in the "no HS" samples by 18 h AHS (Fig. 1C). In *wild type* differentiation, Bam is most highly expressed in the 4cc and 8cc (McKearin and Ohlstein, 1995), whereas the burst of Bam induced by the heat shock persists for a shorter time. To monitor the variation in Bam induction between cells that is not captured by western blot analyses, we used IF, which revealed that the loss of Bam protein is variable at the cellular resolution: at 9 h AHS, we observed a wider range of Bam intensity between differentiating cysts in comparison with 0.5 h or 5 h AHS (Fig. 1D). While the basis of such variability has not been determined, it is possible that this explains the increasing heterogeneity of cyst development we observed by 38 h AHS (Fig. 1B).

To examine the progression of egg chamber development beyond terminal differentiation, we examined the expression of Orb, which is normally concentrated into the future oocyte during the 16-cell cyst stage (Lantz et al, 1994). IF revealed that Orb does not concentrate into a single cell of the 16cc by 38 h AHS (Fig. 1E), suggesting that this final time point captures the final stages of differentiation, prior to egg chamber formation. At 5 days AHS most egg chambers contain a single Orb-positive cell. We also stained for C(3)G to visualise the formation of the synaptonemal complex (Page and Hawley, 2001). We found that C(3)G is lowly expressed at 38 h AHS, with synaptonemal complex being observed at 48 h AHS (Fig. 1F). Finally, as previously shown (Ohlstein and McKearin, 1997), females treated with our heat shock protocol laid eggs that hatched and produced adult offspring. These findings confirm that normal germline development can be fully recapitulated in our synchronisation system.

## Transcriptome measurements in synchronised differentiating GSCs reflect *wild type* differentiation

To examine mRNA expression during differentiation in synchronised GSCs, we performed RNA-seq on rRNA-depleted RNA extracted from $bam^{-/-}$,*hs-bam* ovaries collected at specified time points AHS (Fig. 1A). We generated RNA-seq libraries from ovaries at 'no HS' (*bam* mutant) and 5 h (CB), 9 h (2cc), 18 h (4cc), 28 h (8cc) and 38 h (16cc) AHS. Comparison between biological replicates showed that the data are highly reproducible ($R^2 > 0.97$).

To validate that the transcriptome measurements in the $bam^{-/-}$,*hs-bam* time course reflect mRNA accumulation during

*wild type* differentiation, we used single-molecule fluorescence in situ hybridisation (smFISH) to visualise mRNA transcripts in *wild type* germaria (Figs. 2A and EV1A). Marking the cell boundaries (phalloidin, F-actin) and the spectrosome and fusome (alpha-spectrin) enabled us to identify all the stages of differentiation in *wild type* ovaries. *vasa* mRNA is detected at ~ 200 FPKM throughout the RNA-seq time course, while *aubergine* (*aub*) mRNA is detected at a lower, but constant, level (~100 FPKM). Accordingly, using smFISH, we observed higher numbers of *vasa* mRNA transcripts in the cytoplasm of germ cells at all differentiation stages (Fig. 2Ai), compared to *aub* mRNA (Fig. 2Aii). *matrimony* (*mtrm*) mRNA presented levels below 10 FPKM until the 28 h AHS time point, with a further increase at 38 h AHS. This pattern of upregulation is also observed in *wild type* differentiation, with *mtrm* mRNA only observed in 8cc and 16cc (Fig. 2Aiii). *Benign gonial cell neoplasm* (*bgcn*) displays the opposite mRNA expression pattern, with declining mRNA levels in both the synchronised differentiating GSCs and *wild type* differentiation (Fig. 2Aiv) (Ohlstein et al, 2000). Notably, the synchronised differentiating GSC dataset recapitulates recently published mRNA expression patterns for *CG32814/eggplant*, which is reported to be downregulated after the GSC (Sun et al, 2023), and *blanks*, which is downregulated at the 8cc stage (Blatt et al, 2021) (Fig. EV1A). Collectively, validations by smFISH indicate that the RNA-seq data in synchronised differentiating GSCs mirrors what is observed in *wild type* ovaries both in terms of mRNA levels and changes during differentiation. The smFISH experiments suggest that an expression level in the RNA-seq below 10 FPKM corresponds to a very small number of mRNA transcripts that is only sporadically detected in cells, so we set this as the threshold for further analysis. In our RNA-seq data, 6532 genes were expressed at >10 FPKM in at least one time point.

With the advent of single-cell sequencing technologies, others have recently used single-cell (sc)RNA-seq analysis to characterise different cell types in adult *Drosophila* ovaries (Jevitt et al, 2020; Rust et al, 2020; Slaidina et al, 2021; Sun et al, 2023). For instance, Rust et al, used pseudotime analysis to identify transcriptional signatures, with 12 "marker genes" representing different stages of germline development. As additional validation of our dataset, we compared these data to our bulk RNA-seq from the synchronised GSCs and found that our dataset is consistent with the pseudotime analysis inferred from scRNA-seq (Fig. EV1B), but with increased sequencing depth.

An important caveat is that the protocol used to drive Bam expression is also expected to induce acute changes in the expression of the heat shock-responsive genes (Pauli et al, 1992). Transcript-level changes of the heat shock-responsive genes are characterised by their transient nature, quickly dampening over short periods after the stimulus is removed. To identify genes with acute changes in mRNA level upon heat shock treatment, we generated RNA-seq libraries from $bam^{-/-}$,*hs-bam* ovaries at 0.5 h AHS. 68 genes underwent a significant >threefold change in transcript level in the 0.5 h AHS time point compared to "no HS" (with >10 FPKM expression in at least one of these time points). Most of these genes recovered towards the "no HS" level during the time course, but at differing rates (Fig. EV1C). Gene ontology analysis showed that this cohort of genes was highly enriched for heat response and protein folding terms (Fig. EV1D). Therefore, this small subset of genes, along with an additional 20 contaminant

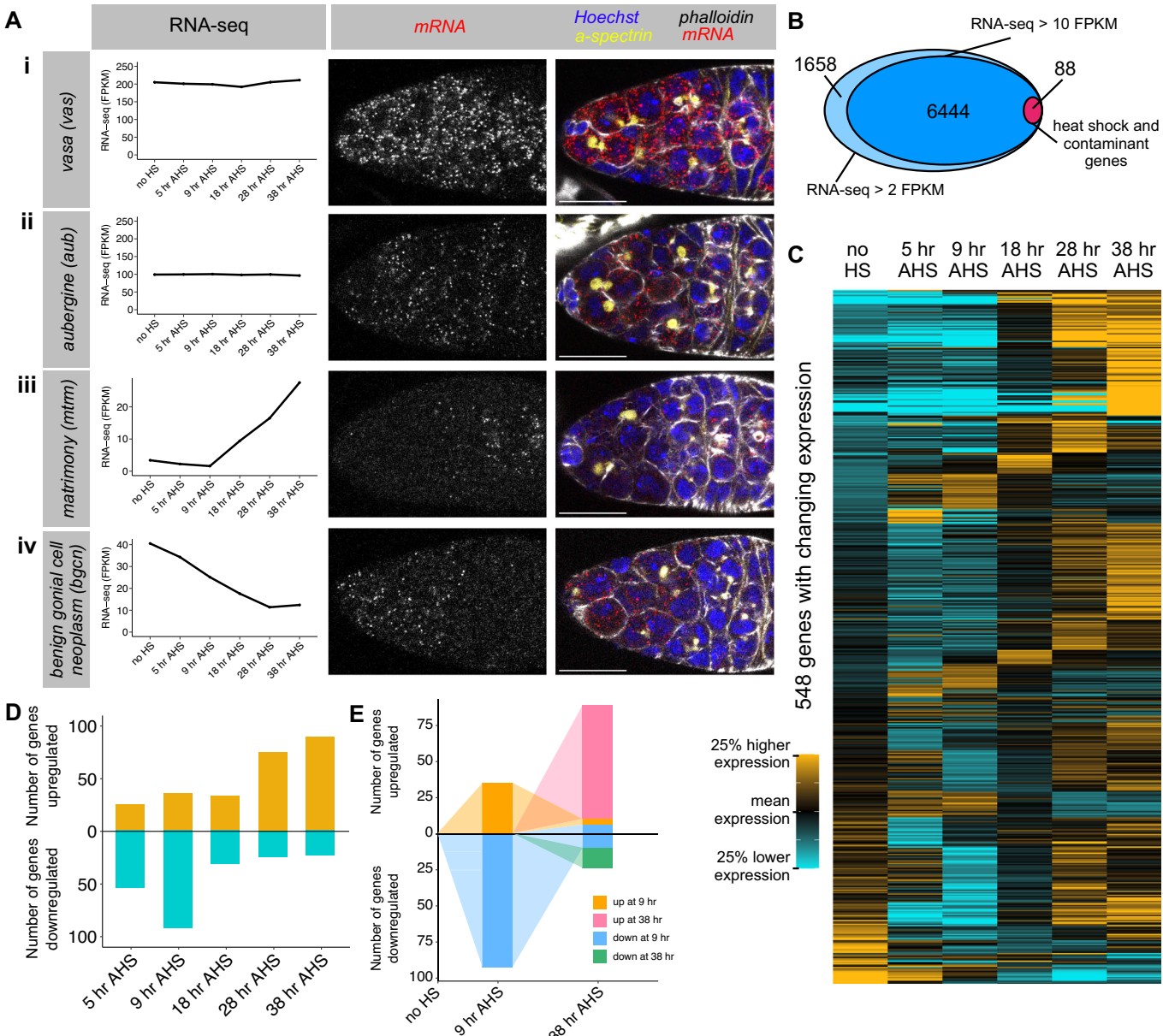

**Figure 2. RNA-seq in synchronised differentiating GSCs reveals dynamic changes in the transcriptome.**

(A) single-molecule fluorescent in situ hybridisation (smFISH) was used in *wild type* germaria (right) to validate the RNA expression measurements in the synchronised differentiating GSC RNA-seq dataset (left). *Wild type* germaria are stained for DNA (Hoechst, blue), actin (phalloidin, gray), fusome (α-spectrin, yellow), and mRNA transcript of interest (smFISH, red and grayscale): (i) *vas*, (ii) *aub*, (iii) *mtrm* and (iv) *bgcn*. Scale bars are 15 μm. (B) RNA-seq in the differentiating GSCs detected 8177 genes expressed at least 2 FPKM at one time point (excluding the 0.5 h heat shock sample), and 6535 of these were expressed to at least 10 FPKM. 90 genes were excluded due to a strong heat shock effect or chorion contamination. (C) Heatmaps illustrating RNA-seq expression level across the time course. Each row represents one gene of the 548 genes that exhibit a significant 1.6-fold change in expression between two time points. Expression level is scaled per gene such that black represents the mean expression across the time course, gold represents a 25% higher expression than the mean, and cyan represents a 25% lower expression than the mean. Genes were grouped by fold change in mRNA expression, starting with the final time point. (D) Number of genes with a significant 1.6-fold change at each time point compared to "no HS". Gold = upregulation, cyan = downregulation. (E) Illustrating the overlap between genes with significant up- or downregulation at 9 h AHS and 38 h AHS. Orange and blue represent genes with changing expression at 9 h AHS, and these are followed to 38 h AHS. Additional genes with changing expression at 38 h AHS are coloured pink and green. Source data are available online for this figure.

genes (encoding proteins of the chorion and vitelline membrane, which are produced by somatic cells), were excluded from downstream analysis (Dataset EV1). This filtering resulted in a dataset of 6444 genes expressed during the time course (Fig. 2B; Dataset EV2).

## GSC differentiation involves two waves of changes in the transcriptome

Having established that the synchronised GSC RNA-seq dataset parallels what is observed during *wild type* differentiation, we

aimed to identify the set of genes for which mRNA levels change during GSC differentiation. To do so, we performed differential expression analysis and found 548 genes (~8.5% of expressed genes) with a significant ($P < 0.05$) and greater than the 1.6-fold difference in mRNA level between any two time points (in which the expression was >10 FPKM in one of the compared time points, and excluding the 0.5 h AHS sample) (Fig. 2C; Dataset EV3). Compared to "no HS", we found that most mRNA downregulation (65%) is observed at the 5 h and 9 h AHS time points (CB and 2cc), while most upregulation (64%) is observed at the 28 h and 38 h AHS time points (8cc and 16cc) (Fig. 2D).

We aimed to identify putative "stem cell-specific" or "differentiation-specific" genes that are differently expressed in the GSC versus differentiating cells. Unexpectedly, only two genes (*CG17127* and *CG14545* (Fig. EV1Av)) were downregulated by at least 1.6-fold from 5 h AHS continually until 38 h AHS. As a "differentiation-specific gene" only *CG11892* was upregulated from 5 h AHS onwards. This finding argues against the presence of a unique, stem cell-specific transcriptional program that must be erased at the onset of differentiation.

Strikingly, we found that most genes (84%) that are up or downregulated at 9 h AHS (2cc) returned to "no HS" levels by 38 h AHS (16cc) (Fig. 2E). In contrast, the majority (82%) of mRNA expression changes at 38 h AHS are contributed by genes which exhibited no significant (>1.6-fold) change at 9 h AHS. These data are inconsistent with a model of accumulating changes from GSC to 16cc, and instead reveal two distinct waves of dynamic changes in the transcriptome throughout differentiation.

To characterise the observed changes further, we asked which genes contribute to each wave of mRNA expression change, and performed gene ontology analysis. We considered genes with mRNA level changes at 9 h AHS (2cc) and at 38 h AHS (16cc) in comparison to "no HS" (Fig. EV2A). We found that the group of genes upregulated at 9 h AHS was enriched for genes involved in DNA replication and the cell cycle (Fig. EV2Bi), while downregulated genes were enriched for genes involved in the regulation of lipid storage. At 38 h AHS the upregulated genes were enriched for annotations of the polar granule (GOterm: P granule) (Fig. EV2Bii), which are assembled during oogenesis (Lehmann, 2016), with no significant enrichment for downregulated genes. At 38 h AHS, the upregulated group includes genes known to play a role in meiosis (*mtrm, c(3)g, orb, corolla*) and oocyte development or early embryogenesis (*osk, bru1, dhd, png, alphaTub67C*), while the downregulated group of genes includes *bgcn, blanks, RpL22-like*, and *eIF4E3*.

## Splicing analysis reveals changes in exon inclusion and promoter usage during differentiation

Differential expression analysis at the gene annotation level obscures changes in mRNA splicing, which can alter the encoded protein isoform. Therefore, we quantitated changes in splicing independently of changes in mRNA expression level by pair-wise analysis of our RNA-seq datasets using JUM (Wang and Rio, 2018). The resulting measurement of deltaPSI (percent spliced in) describes the change in the usage of each splicing event between two conditions. Overall, we observed a large number of splicing changes, with different genes changing in splicing in the early wave (9 h AHS) compared to the late wave (38 h AHS) (Fig. 3A;

Dataset EV4). Changes in splicing are reflected in the upregulation of one splice site (on the right of the graphs) in exchange for the downregulation of an alternative site (on the left of the graphs).

Our analysis identified both known and novel examples of splice isoform changes that modify the encoded proteins during differentiation. Over-expression of isoform-specific cDNA constructs has been used previously to infer that a burst of cytoplasmic Rbfox1 during late GSC differentiation is caused by a change in splice isoform to exclude the nuclear localisation signal (*Rbfox1-RN/F* isoform annotated by FlyBase) (Carreira-Rosario et al, 2016; Tastan et al, 2010). In our GSC differentiation time course, this alternative splicing event is captured and is most pronounced at 28 h AHS (Fig. 3B). Hu li tai shao (Hts) is a key component of the fusome, and is required for coordinating cyst division and oocyte specification (Yue and Spradling, 1992). *hts* does not show a significant change in mRNA expression level during our GSC differentiation time course, but alternative splicing analysis shows a change in splicing at the 3' end of the transcript, producing alternative protein isoforms. During differentiation, the ratio of 3' end usage shifts such that *hts-RO/R/S/Q/L/M* (*adducin*, encoding a conserved F-actin and Spectrin-binding protein) and *hts-RA/K/N* (*ovhts*) are increased at 38 h AHS compared to "no HS" (Fig. 3C) (Whittaker et al, 1999; Gerdes et al, 2020). The encoded Ovhts polyprotein is cleaved to release HtsRC, which specifically localises to the ring canals during cyst development, therefore our data shows that HtsRC production is upregulated at the level of alternative splicing. The mRNA isoform *hts-RP* was thought to be testis-specific (Gerdes et al, 2020) but is quite highly expressed in the "no HS" sample and is downregulated during our time course. Hence, developmentally regulated alternative splicing can lead to functional changes to the protein isoforms expressed in the cell without a change in overall gene expression level, revealing a layer of regulation during GSC differentiation that has not yet been systematically explored.

Unexpectedly, splicing analysis also provided insight into the mechanism of regulation of genes that exhibit changing expression level during differentiation. We found that *bruno* (*bru1*), encoding an RNA-binding protein (RBP) that functions to inhibit the expression of the mitotic cyclins after meiotic entry (Sugimura and Lilly, 2006), is expressed in the GSC and upregulated during differentiation, but that this regulation occurs through the activation of an additional downstream transcription start site (TSS) (Fig. 3D). In the "no HS" sample, the majority of transcription initiates at the most 5' start site (producing the *bru1-RK* transcript), while at 38 h AHS, a downstream TSS is also active, producing the *bru1-RA/E* transcripts, which likely underlies the upregulation of Bru1 protein observed during differentiation. Similarly, *Rbp9*, encoding another RBP required for proper cyst differentiation and oocyte determination (Kim-Ha et al, 1999), is also upregulated during differentiation. The splicing analysis uncovered the activation of an additional TSS to express the *Rbp9-RB/E/G/J* transcripts at 38 h AHS (Fig. 3E). In the examples of *bru1* and *Rbp9*, changes in TSS usage lead to the use of different 5' untranslated sequences (5' UTR) without a change in coding sequence. Lesser-studied genes include *bip1*, which is also upregulated during GSC differentiation and shows specific increased transcription from an upstream TSS at 38 h AHS to produce *bip1-RA* (Fig. 3F). In this case, differential TSS usage is expected to lead to the expression of different protein isoforms.

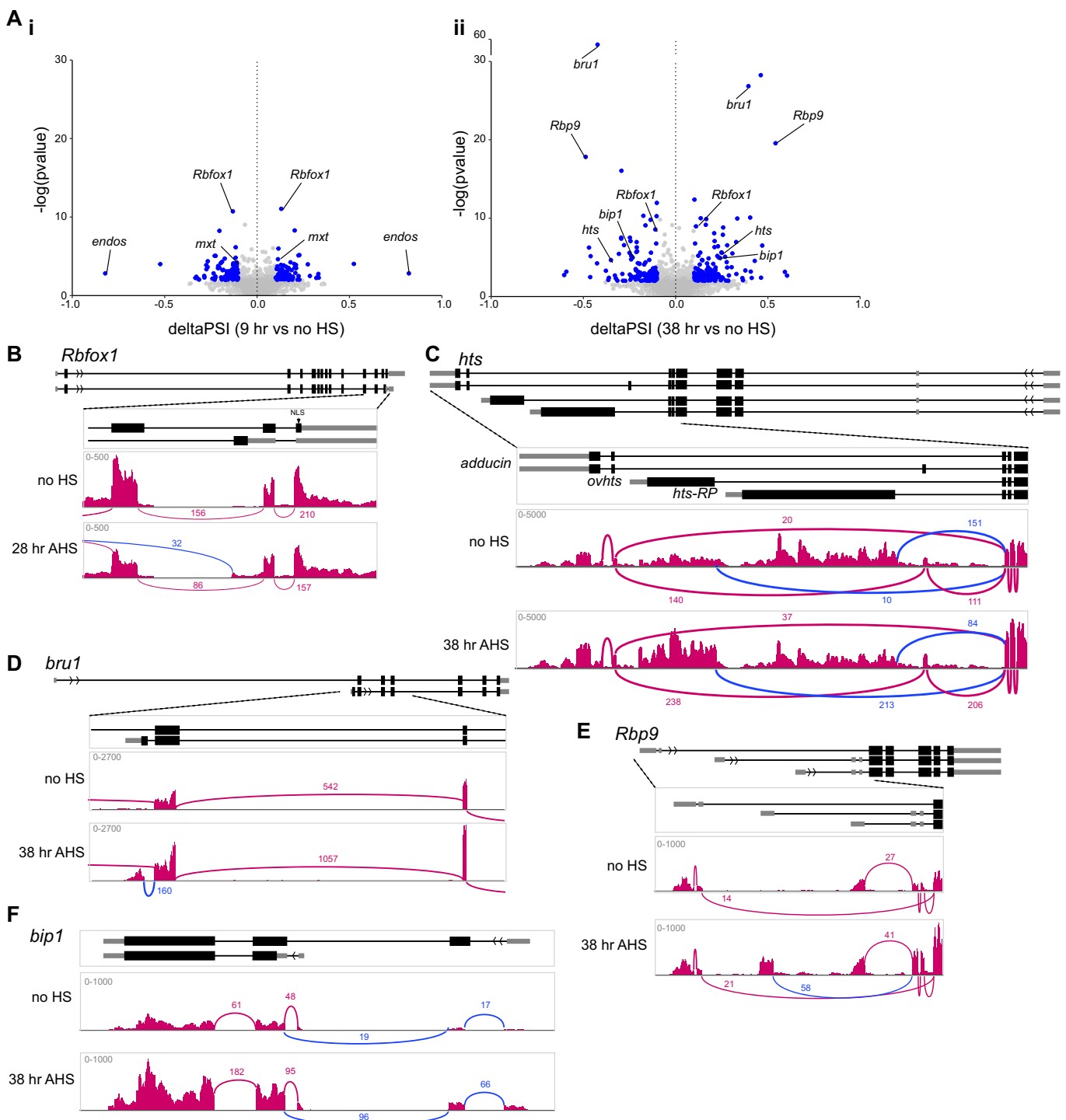

**Figure 3. Changes in splicing during differentiation can alter regulation or protein isoform.**

(A) Splice site usage at 9 h AHS (i) or 38 h AHS (ii) compared to "no HS", plotted as deltaPSI (percent spliced in), calculated using JUM (Wang and Rio, 2018). Blue points meet the thresholds of >0.1 or <−0.1 deltaPSI and P < 0.01. (B–F) Example changes in splicing of *Rbfox1* (B) and *hts* (C), as well as transcription start site usage of *bru1* (D), *Rbp9* (E), and *bip1* (F). The simplified gene structure is shown above, and RNA-seq read tracks from IGV are shown with Sashimi plots to visualise splice junctions. Changing splicing of interest is highlighted in blue.

Altogether, these examples illustrate how activation of an alternative TSS is a mechanism to upregulate gene expression.

## Ribo-seq captures translation dynamics during GSC differentiation

RNA-seq measures gene expression changes at the mRNA level, which are often thought of as a proxy for consequential changes at the protein and cell fate level. Indeed, scRNA-seq has been broadly used as a readout for fate changes (Morris, 2019; Brunet Avalos et al, 2019; Schiebinger et al, 2019; Farrell et al, 2018). This premise assumes no change in the translation efficiency of the transcripts, but mounting evidence indicates that transcripts for different genes are not equally translated (Vogel and Marcotte, 2012). Translation regulation plays a significant role during GSC differentiation—indeed, the master differentiation factor Bam is itself a regulator of translation (Li et al, 2009; Slaidina and Lehmann, 2014). Despite this, the dynamics of translation changes during stem cell differentiation are unknown, largely due to the large amount of material required to analyse translation genome-wide. To circumvent these difficulties, we applied the $bam^{-/-}$,hs-bam system to obtain enough starting material to perform Ribo-seq at the same time intervals as the RNA-seq dataset. Even using the synchronisation approach, the amount of tissue was limited compared to previous Ribo-seq experiments in *Drosophila* S2 cells (Aspden et al, 2014; Douka et al, 2022), embryos (Dunn et al, 2013), oocytes (Kronja et al, 2014) or ovaries (Jang et al, 2021), so we first performed quality control analysis on the Ribo-seq libraries (Fig. EV3). Comparisons between biological replicates revealed that the data are highly reproducible ($R^2 > 0.97$). As expected from high-quality Ribo-seq data, we observed that the majority of the reads (58–70%) showed the expected ribosome footprint (28–32-nt long), overwhelmingly mapped to the sense direction of the CDS (>90%) or 5' UTR (~8.9%) and showed a strong 3-nucleotide P-site periodicity. This benchmarking confirms that our Ribo-seq is of high-quality, despite the limitations in input material.

We assessed the translation of the 6444 genes that we had determined to be expressed by RNA-seq (Fig. 4A; Dataset EV5). Of these, 43 further genes showed heat shock effects in the Ribo-seq (>threefold change in the 0.5 h AHS sample compared to "no HS"). Using a threshold of 10 FPKM in the Ribo-seq, we found that 5922 of the expressed genes were translated at some point during our time course, while 479 genes were not translated at any time point (<10 FPKM).

Just as we validated the RNA-seq datasets against *wild type* differentiation using smFISH, we aimed to validate the Ribo-seq through independent approaches. We first compared the Ribo-seq reads for *bam* to the results we obtained by western blot (Fig. 1C). From the Ribo-seq, we found that *bam* translation peaks at 0.5 h AHS, but declines gradually (Fig. 4B), which matches the western blot. We then focused on hallmark genes that have been previously characterised as being translationally regulated during GSC differentiation. For instance, *osk* mRNA, which is crucial for the assembly of the germplasm at the posterior pole of oocytes, is not transcribed until the late stages of GSC differentiation, but remains translationally repressed in differentiated cells until much later in oogenesis (Ephrussi et al, 1991; Kim-Ha et al, 1991, 1995). In agreement, the RNA-seq data revealed a progressive increase in the

level of *osk* mRNA from 18 to 38 h AHS (4–16cc), while the Ribo-seq showed that *osk* mRNA was not engaged with ribosomes throughout the time course (Fig. 4Ci). On the other hand, transcriptional reporters suggested that the *nanos* (*nos*) gene is transcribed throughout GSC differentiation, while the protein level decreases during the CB to 4cc stage (Li et al, 2009). The Ribo-seq data revealed a sharp decrease in *nos* translation at the 5 h and 9 h AHS time points (CB to 2cc), but surprisingly, the mRNA level mirrored this change, which together with constant transcription implies decreased mRNA stability (Fig. 4Cii). Both the RNA-seq and Ribo-seq showed full recovery of *nos* by 28 h AHS (8cc).

While our datasets recapitulate examples of translation regulation already shown in the literature, we wanted to validate the Ribo-seq in an unbiased manner. To do so, we selected genes with similar RNA expression levels but different Ribo-seq read levels and procured FlyFOS GFP-tagged transgenic lines. The FlyFOS transgenes are expected to recapitulate the expression of the endogenous gene, as they were designed with the goal to include all regulatory sequences existing on endogenous genomic loci (Sarov et al, 2016). We stained germaria with *gfp* smFISH to measure RNA level, and GFP protein to measure the level of each fusion protein (Fig. 4D). Using the same *gfp* smFISH probes and anti-GFP antibody minimises confounding detection effects. In agreement with the RNA-seq data, the smFISH showed similar numbers of mRNA transcripts for each gene. Moreover, mirroring the results from the Ribo-seq analysis, the GFP protein levels were very different for each gene (highest = *mod*, medium = *Dp1*, lowest = *Cirl*). Therefore, we conclude that the Ribo-seq in synchronised differentiating GSCs reflects changes in translation during *wild type* differentiation. Ribo-seq data reports the rate of protein production during differentiation, providing more direct insight into the changes in protein complement during differentiation compared with RNA-seq.

## Ribo-seq identifies two waves of global remodelling of translation during GSC differentiation

We asked if changes in mRNA expression corresponded to changes in the Ribo-seq. 548 genes showed a >1.6-fold change in RNA level between any two time points, and 353 of these (64%) also showed a significant 1.6-fold change between two samples in the Ribo-seq dataset (Fig. 4A). However, an additional 1360 genes showed a change in the Ribo-seq but not the RNA-seq data, revealing that these genes are regulated primarily at the level of translation. In total, 8.5% of transcribed genes show a >1.6-fold change in RNA expression at some point over the time course, while 29% of translated genes show a >1.6-fold change in the Ribo-seq experiment. This result illustrates that a more extensive remodelling of the translatome than the transcriptome is at play during GSC differentiation.

In total, 1713 genes showed a >1.6-fold difference in translation level between any two time points (excluding the 0.5 h AHS sample) (Fig. 5A; Dataset EV6). Comparing the 38 h AHS time point to "no HS", 99 genes were upregulated, including *orb, grk, moon, png, c(2)M, bru1, alphaTub67C* and *Rbp9*, while 165 were downregulated at the level of translation, including *bgcn, stau* and *Pxt*. There were roughly twice as many downregulated genes as upregulated genes at each time point, with two waves of changes

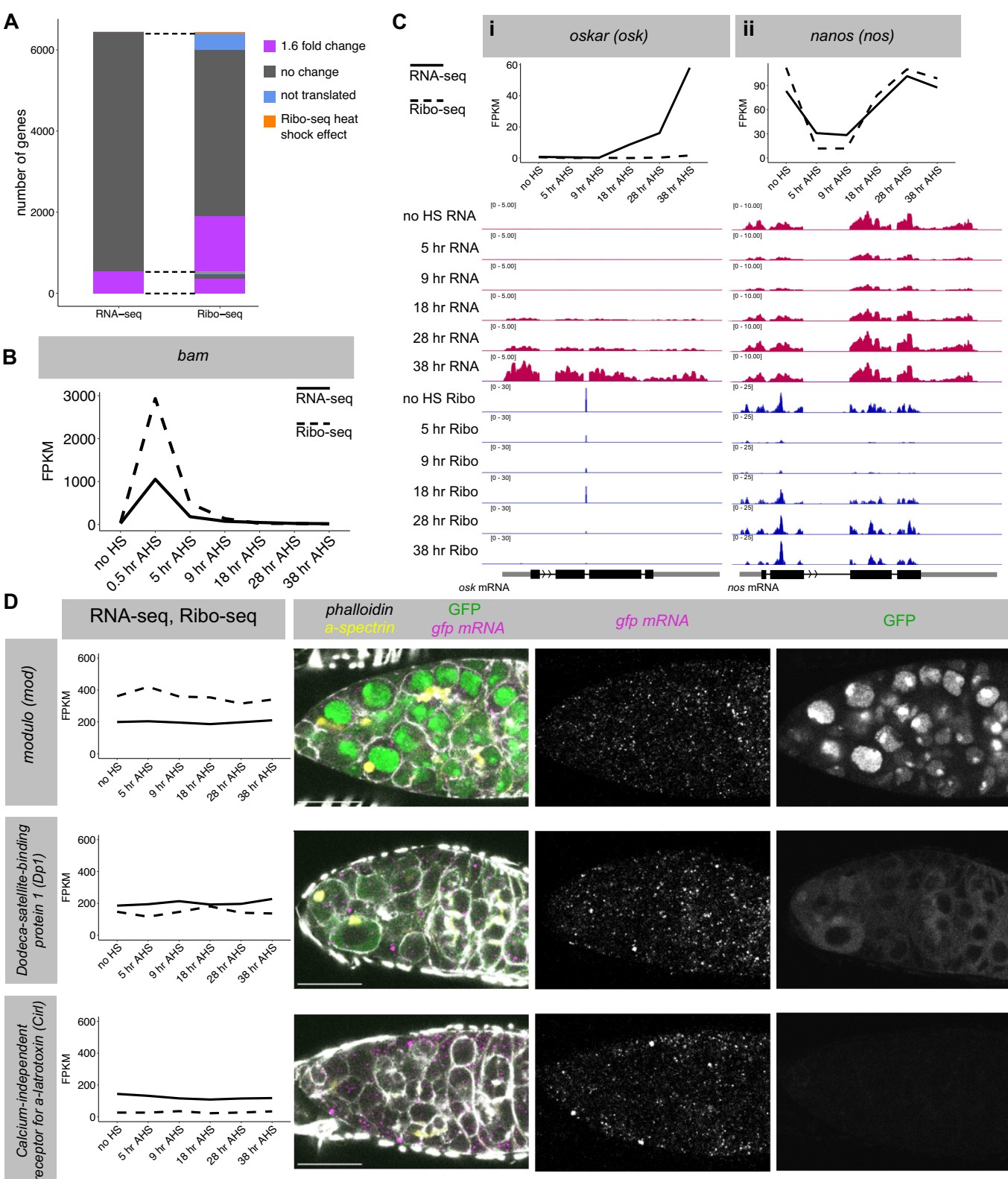

around an inflection point at 18 h AHS (Fig. 5B). Similarly to the RNA-seq analysis, we found that the majority (88%) of genes that increase or decrease in translation in the early wave (9 h AHS, 2cc), returned to "no HS" levels by 38 h AHS (16cc) (Fig. 5C). The majority (77%) of translation changes in the late wave are contributed by genes that exhibited no change in the early wave. At the 18-h inflection point, some key changes are observed, including the downregulation of *bgcn* and *staufen*, and the initial

**Figure 4. Ribo-seq in synchronised differentiating GSCs recapitulates changes in translation during normal differentiation.**

(A) After filtering, 6444 genes are expressed at >10 FPKM in at least one time point by RNA-seq. In total, 548 of these genes show a >1.6-fold change between any two samples. Of these 548 genes, 353 also show a 1.6-fold change in Ribo-seq reads between two samples. 1360 genes show a >1.6-fold change in the Ribo-seq with no significant change in the RNA-seq. (B) *bam* transcript level and translation during the time course from RNA-seq and Ribo-seq data (FPKM). (C) RNA-seq and Ribo-seq in the synchronised GSC system recapitulate previously characterised regulation of *osk* (i) and *nanos* (ii). IGV tracks for each time point for RNA-seq (red) and Ribo-seq (blue). (D) GFP-tagged FlyFOS constructs (*modulo, dodeca satellite-binding protein 1, calcium-independent receptor for a-latrotoxin*) stained for *gfp* smFISH (magenta and individual greyscale) and GFP protein (green, and individual greyscale), f-actin (phalloidin, greyscale) and fusome (α-spectrin, yellow). Scale bars are 15 μm. graphs: solid line = RNA-seq, dashed line = Ribo-seq. Source data are available online for this figure.

upregulation of some genes of the second wave, including *alphaTub67C* and *dhd*.

We performed gene ontology analysis and found that, as in the RNA-seq, the first wave of upregulation was enriched for genes involved in cell cycle and DNA replication, as well as cytoskeleton organisation (Fig. EV4A,B). The early wave of downregulation in the Ribo-seq was enriched for genes involved in translation, primarily the ribosomal proteins, suggesting that the previously observed downregulation of ribosome biogenesis during GSC differentiation (Neumüller et al, 2008; Zhang et al, 2014) is itself regulated at the level of translation. There was no significant enrichment in the second wave at the level of translation. Indeed, many of the upregulated transcripts that are enriched in the polar granules are translationally repressed at this stage, as for *osk*.

To get a global perspective of the changes occurring at the transcriptome and translatome level, we used Principal Component Analysis (PCA) to examine the variance in the RNA-seq and Ribo-seq samples throughout our time course (Fig. 5D). In the RNA-seq, PC1 explains ~37% of the variance, in which the 9 h AHS and 38 h AHS samples are the most separated. Illustrating an inflection point around 18 h AHS, this sample was the most similar to the "no HS" sample. PC2 explains ~28% of the variance and groups the mid samples (9 h AHS, 18 h AHS and 28 h AHS) closer together. In the Ribo-seq samples, PC1 explains ~34% of the variance and again shows the biggest difference between 9 h AHS and 38 h AHS, with "no HS" and 18 h AHS positioned close to each other. Thus, globally, our data reveal that both the transcriptome and translatome undergo two waves of remodelling during differentiation, with a stage in between in which differentiating cells present an expression program that is remarkably close to that of the stem cells.

## Transcripts that are translationally repressed during differentiation are often translated later during oocyte or embryo development

A cohort of genes are expressed but translationally repressed throughout our time course, so we asked whether they are translated later in development rather than reflecting spurious transcription. To explore this hypothesis, we examined the expression and translation of these genes in previously published RNA-seq and Ribo-seq experiments using the whole ovary as input (Greenblatt and Spradling, 2018) (Data ref: NCBI BioProject PRJNA466150 (2018)). Whole ovary datasets primarily represent expression and translation in maturing egg chambers, which make up the bulk of the tissue. We focussed on two groups of untranslated genes from our time course (<10 FPKM in the Ribo-seq throughout the time course): first the "upregulated" group

(>1.6-fold increase in RNA level from "no HS" to 38 h AHS, 30 genes, Fig. 6A, blue points), and secondly the "constitutive" group (>15 FPKM at every time point of the RNA-seq, 129 genes, Fig. 6A, purple points).

Analysis of the mRNA expression and translation of the "upregulated" group (30 genes) revealed that 12 of these genes were expressed to >10 FPKM in the whole ovary RNA-seq, and of those, 8 were translated to >10 FPKM in the whole ovary Ribo-seq (Fig. 6B, blue points). This group includes *osk*, which has been shown to be loaded into the developing oocyte and translated at the posterior pole (Ephrussi et al, 1991; Kim-Ha et al, 1991, 1995), as well as meiosis genes (including *dhd, mtrm* (Fig. 6Ci) (Xiang et al, 2007) and *wisp*), and *yolkless* (*yl*), which is involved in vitellogenesis of the egg (DiMario and Mahowald, 1987).

Of the "constitutive" group of genes, 90 were expressed to >10 FPKM in the whole ovary RNA-seq (Fig. 6B, purple points). In all, 30 of these were also translated to >10 FPKM in the whole ovary Ribo-seq, including *giant nuclei* (*gnu*), which regulates the PNG kinase complex driving the early embryonic nuclear divisions (Hara et al, 2017). The remaining 60 genes remain translationally repressed, including the RBP-encoding gene *smaug* (*smg*), which is known to be loaded as mRNA into the oocyte and is translationally repressed until egg activation, downstream of the PNG kinase (Tadros et al, 2007). Our data show that *smg* mRNA is expressed throughout early GSC differentiation (Fig. 6Cii), and so the translation repression mechanism must be active throughout differentiation and oogenesis. In this case, a robust translational repression mechanism may make any additional transcriptional regulation obsolete.

We hypothesised that some genes that are translationally repressed during GSC differentiation but not expressed in the whole ovary RNA-seq, might be translated at a specific stage of egg chamber development but fall beneath the detection threshold in the whole ovary datasets. Indeed several genes expressed only during the first stages of differentiation in our dataset (e.g., *blanks, bgcn*), are not detected >10 FPKM in the whole ovary RNA-seq. For example, *mei-218*, which encodes a protein involved in meiotic recombination (Manheim et al, 2002), is not translated in our GSC differentiation time course, and does not meet the detection threshold in the whole ovary RNA-seq. We performed smFISH and showed that *mei-218* mRNA is expressed throughout the germarium and upregulated at the 16cc stage (Fig. 6Ciii). *mei-218* mRNA is then downregulated but continues to be lowly expressed in developing egg chambers with enrichment in the oocyte. This is consistent with published in situ and antibody stainings showing that Mei-218 protein is upregulated in the 16cc during meiotic prophase (Manheim et al, 2002). These examples illustrate that translational repression is implemented widely during GSC differentiation to prepare the pool of transcripts in advance of a change of fate during development.

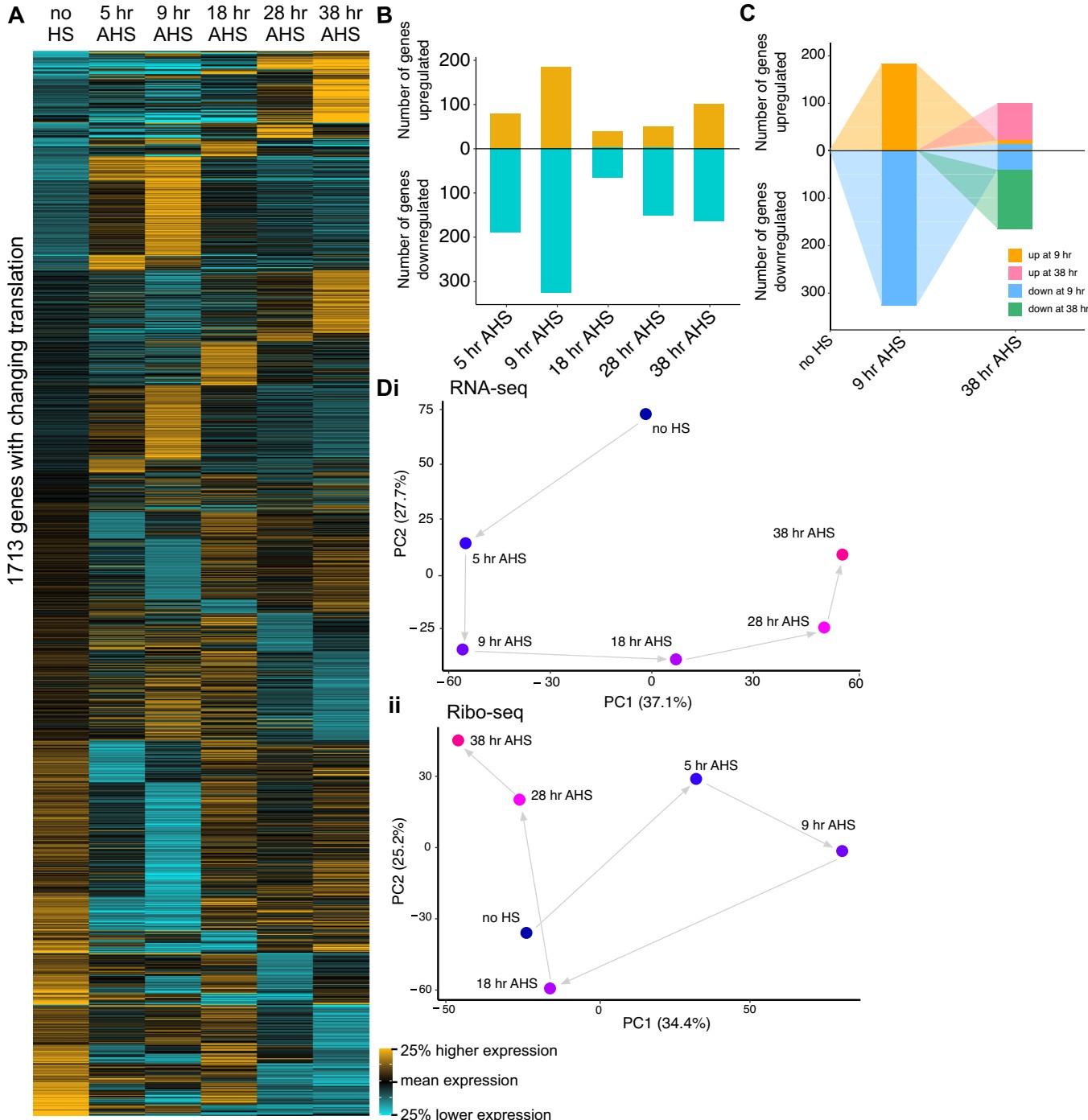

**Figure 5. Ribo-seq in synchronised differentiating GSCs uncovers changes in translation during differentiation.**

(A) Heatmap illustrating translation level across the time course. Each row represents one gene of the 1713 genes that exhibit a significant 1.6-fold change in translation between any two time points. Expression level is scaled per gene such that black represents the mean translation (Ribo-seq FPKM) across the time course, gold represents a 25% higher translation than the mean, and cyan represents a 25% lower translation than the mean. Genes were grouped by fold change in Ribo-seq, starting with the final time point. (B) Number of >1.6-fold changes in Ribo-seq at each time point compared to "no HS". Gold = upregulation, Cyan = downregulation. (C) Illustrating the overlap between genes with significant up- or downregulation in the Ribo-seq at 9 h AHS and 38 h AHS. Orange and blue represent genes with changing expression at 9 h AHS, and these are followed to 38 h AHS. Additional genes with changing expression at 38 h AHS are coloured pink and green. (D) PCA of the RNA-seq (i) and Ribo-seq (ii) in the differentiating GSC time course.

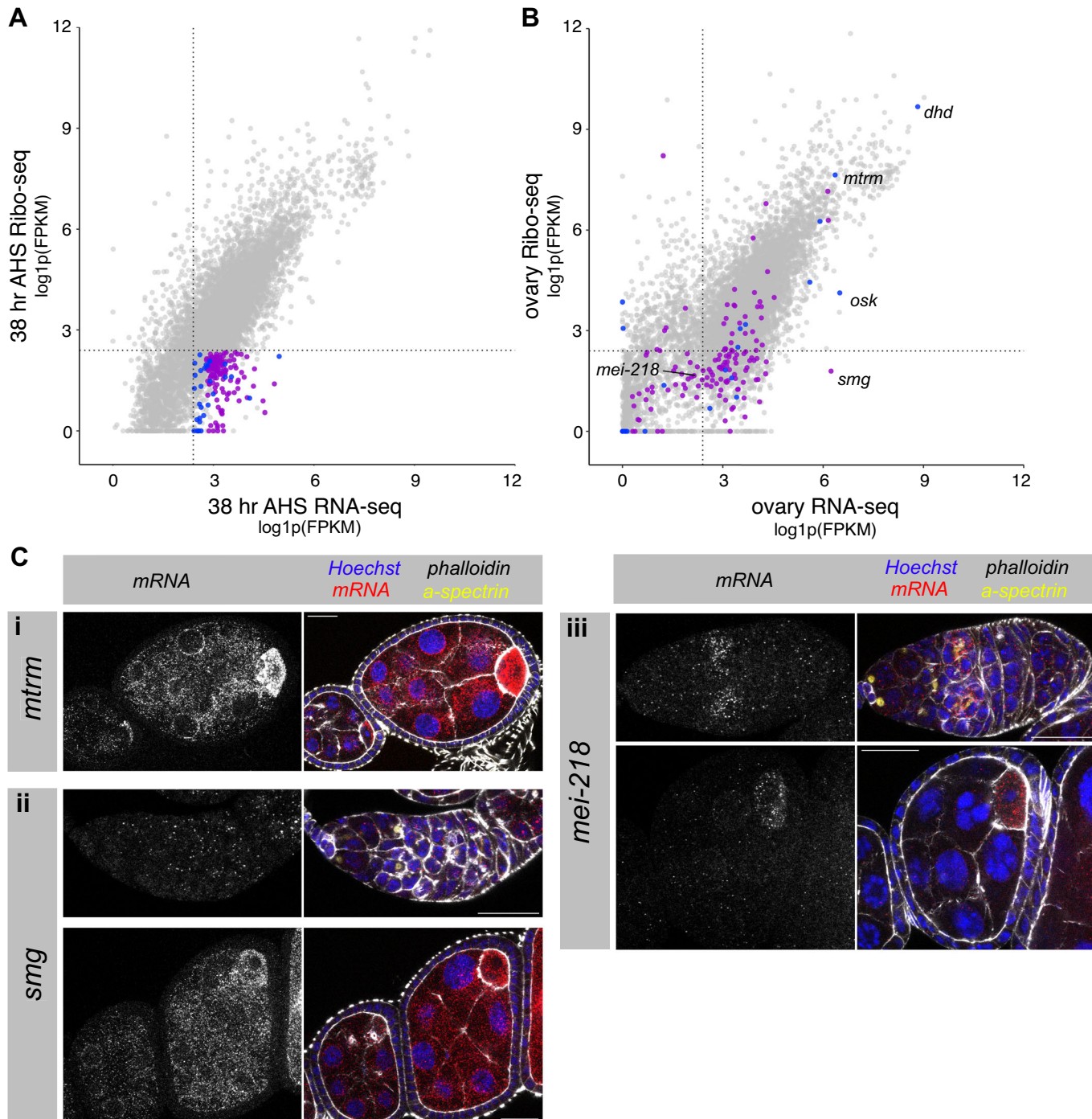

**Figure 6. Translationally repressed genes during differentiation are often translated later in the egg chamber or embryo development.**

(**A**) FPKMs for each gene in the RNA-seq and Ribo-seq datasets at 38 h AHS, plotted as log1p(FPKM). Highlighted genes are untranslated (<10 FPKM throughout the time course). Blue represents upregulated genes (>1.6-fold increase in RNA level from "no HS" to 38 h AHS), purple represents constitutive genes (>15 FPKM in the RNA-seq at every point of the time course). Dotted lines represent 10 FPKM. (**B**) Plotting log1p(FPKM) in whole ovary RNA-seq and Ribo-seq (Greenblatt and Spradling, 2018). Genes coloured as in (**A**). Dotted lines represent 10 FPKM. (**C**) smFISH in *wild type* ovaries for (i) *mtrm*, (ii) *smg* and (iii) *mei-218. wild type* ovarioles are stained for DNA (Hoechst, blue), actin (phalloidin, gray), fusome (α-spectrin, yellow), and mRNA transcript of interest (smFISH, red and grayscale). Scale bars are 20 µm.

## Transcript-level changes can be intensified or buffered at the level of translation through regulation of translation efficiency

Accumulating evidence indicates that changes in the translatome are a composite of differences in mRNA level and differences in translation efficiency (TE) of each gene. TE is a measure that is proportional to ribosomes per transcript, and can be calculated for each gene as Ribo-seq reads divided by RNA-seq reads. When considering individual examples in our dataset, we find cases in which the changes in translation reflect similar changes in mRNA level, e.g. *orb* and *bgcn*, but for others the Ribo-seq is not paralleled

by the RNA-seq (e.g. *osk* and *dhd*), implying that the latter group is regulated at the level of TE (Figs. 4C and 7A). In the case of *corona* (*cona*), a gene encoding a component of the synaptonemal complex required for chromosome pairing, both transcription and translation decrease during differentiation, but the effect is much greater at the level of translation. To systematically and quantitatively determine the changes in TE for each gene, we applied deltaTE (Chothani et al, 2019) on RNA-seq and Ribo-seq data throughout our time course compared to "no HS". With a threshold of >1.6-fold change in TE (with *P* value <0.05 and FPKM of at least 10 in one of the compared RNA-seq samples), we identified 874 genes with changing TE during the time course (Fig. EV5; Dataset EV7).

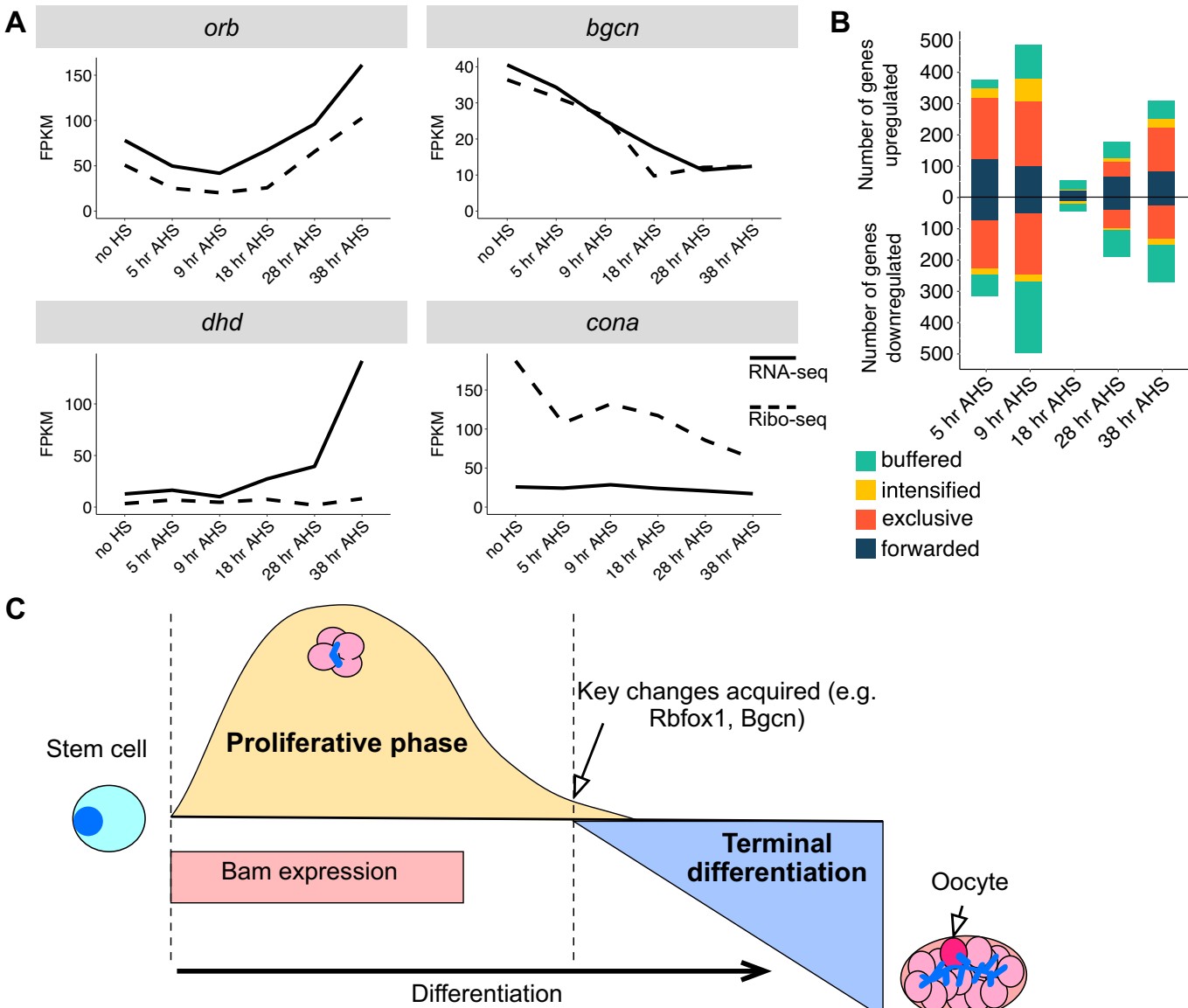

**Figure 7. Transcript-level changes can be intensified or buffered at the level of translation through the regulation of translation efficiency.**

(A) Changing expression of different genes in the RNA-seq (solid line) or Ribo-seq (dashed line). Ribo-seq in *orb* and *bgcn* reflects the RNA-seq, but for *dhd* and *cona* there is additional regulation at the level of translational efficiency. (B) Number of genes classified into each of four regulatory groups (forwarded, exclusive, intensified or buffered) at each time point compared to "no HS". Up- or downregulation is classified by the direction of change in Ribo-seq. (C) Model showing two waves of expression change during GSC differentiation.

To understand the different types of gene regulation in more detail, we grouped genes into four classes (Chothani et al, 2019): regulation only at the mRNA level (forwarded), only at the level of translation efficiency (exclusive), or by mRNA and translation efficiency changing in the same direction (intensified) or opposite directions (buffered). We classified gene regulation into these four categories at each time point relative to the "no HS" sample (Dataset EV8). Overall, we found that translation regulation contributed to the regulation of more genes than mRNA level, with the exception of the 18 h AHS time point where 96% of regulatory events involved changing mRNA level (Fig. 7B). Although "intensified" represents the most effective way to change protein expression, at each time point, only 4–10% of regulatory events are classified as such. Perhaps this finding reflects the complexity of establishing mechanisms for both transcriptional and translational regulation on any given gene. On the other hand, the 'buffered' category—in which translation control neutralises the changes in mRNA levels—was much larger, at each time point making up 14–56% of regulatory events. In these cases, changing TE maintains the status quo at the level of translation, while the transcriptome is remodelled, perhaps in preparation for the coming stages of differentiation. These results emphasise that changes in mRNA level are a less consequential output than measurements obtained by Ribo-seq, and that translation efficiency is a major lever to regulate protein expression during germline stem cell fate transitions.

## Discussion

During differentiation, stem cell progeny undergo a fate change that can transform cellular morphology, metabolism and function, but how regulated gene expression programs coordinate differentiation has been obscured by technical challenges. We have developed a protocol to synchronise GSC differentiation in vivo, allowing us to collect tissue samples to perform RNA-seq and Ribo-seq at high developmental resolution through differentiation. This dataset provides a rich insight into the remodelling of the transcriptome and translatome during cell fate change and unveils two waves of global gene expression changes. Surprisingly, between these waves, the gene expression pattern resolves to a stem cell-like state with only a small number of crucial changes.

The RNA-seq data of synchronised differentiating GSCs was validated through imaging approaches, and reproduces the gene expression patterns identified through scRNA-seq (Jevitt et al, 2020; Rust et al, 2020; Slaidina et al, 2021; Sun et al, 2023). However, the increased depth of the data provided by RNA-seq allowed us to identify more differentially expressed genes in our dataset, which was instrumental to uncover the two wave program. Importantly, we found that the translatome undergoes threefold more changes than the transcriptome during GSC differentiation, which could explain why it has been difficult to specifically distinguish signatures for each stage of differentiation from scRNA-seq data. While the role for translation control is not surprising, given that translation regulators have been shown to be drivers of GSC differentiation (Li et al, 2009; Carreira-Rosario et al, 2016; Teixeira and Lehmann, 2019), the magnitude of the effect was not anticipated. Importantly, we found that changes in mRNA level were not predictive of changes in translation: just a third of the observed changes in the RNA-seq also show a concomitant change in the Ribo-seq. This finding is in line with previous reports that transcript level (as measured by RNA-seq) is not a good proxy for protein level (Vogel and Marcotte, 2012). RNA-seq enables insight into the regulatory processes upstream of protein production, but on its own provides limited explanation of the cell biological changes happening during cell fate transitions.

We analysed how mRNA level and translation efficiency each contribute to changes in translation during differentiation. Changing both the mRNA level and translation efficiency in the same direction has an amplifying effect, but this "intensified" category made up a minority of regulatory events, perhaps because multiple regulatory mechanisms would be superfluous. When both mRNA level and translation efficiency acted on the same gene, they most frequently acted in the opposite direction (buffering the level of translation). The buffered category includes genes which are upregulated at the mRNA level but remain translationally repressed, enabling a preparatory transformation of the transcriptome without immediately affecting the protein complement of the cell. Indeed, we found that many of these genes are translated later in oogenesis or embryonic development. Alternatively, in some cases buffering may be a mechanism of modulating the translation efficiency to enact a more precise regulation than by changing mRNA level alone.

At both the level of transcription and translation, we observed two distinct waves of gene expression changes (Fig. 7C): at 5–9 h and 28–38 h AHS there were many changes relative to "no HS", while 18 h AHS is most similar to the "no HS" undifferentiated sample, indicating an inflection point. Interestingly, the genes changing in each wave had minimal overlap, suggesting that differentiation to a new fate involves two separate phases rather than by the gradual accumulation of changes. Supporting this idea, we did not observe a large cohort of "stem cell-specific genes". Only two genes were downregulated continually during differentiation at the level of transcription, and only six at the level of translation. Many of these have not been characterised, but they included the transcription factor Trf5. Instead, most changes were observed in two distinct waves during differentiation. Early in differentiation, there was an upregulation of genes involved in the cell cycle and DNA replication, which may drive the four mitotic divisions that occur during differentiation. Later in differentiation, this group of genes are no longer upregulated. It has been previously proposed that when the RNA-binding protein Bruno (Bru1) is expressed late in differentiation, it translationally represses mitotic factors leading to exit from the cell cycle (Sugimura and Lilly, 2006; Parisi et al, 2001; Wang and Lin, 2007). In contrast, our findings suggest that many cell cycle factors are regulated primarily at the mRNA level, not at the translation level. During early differentiation, the downregulated genes in the Ribo-seq were enriched for terms involving the ribosome, regulated by decreased TE. This finding is consistent with previous literature that ribosome biogenesis is downregulated during differentiation (Neumüller et al, 2008; Zhang et al, 2014; Martin et al, 2022) and suggests that ribosome biogenesis is regulated at the level of translation. At late stages of differentiation, the transcriptome showed an upregulation of genes enriched in polar granules. This was not reflected in the Ribo-seq, likely because many of these genes were shown to be translationally repressed at the early stages of oogenesis (Lehmann, 2016).

Between the two waves, the inflection point at 18 h AHS shows surprisingly few differences to the GSC-like "no HS" sample, but has a very different forward trajectory of gene expression changes (Fig. 7C). The first wave is driven by the introduction of Bam protein via the heat shock, and Bam levels have resolved near to background levels by 18 h AHS. However, the cells at 18 h AHS do not return to a stable GSC-like fate, and instead embark on the second phase of differentiation without further extrinsic stimulus. Intriguingly, it has previously been shown that 4cc and 8cc stage differentiating female germ cells can dedifferentiate back to GSCs (Kai and Spradling, 2004), perhaps reflecting the inflection point before terminal differentiation. Of note, the timing of the first wave may be shortened in the synchronisation system because Bam persists for a shorter time compared to *wild type* differentiation.

We hypothesise that changes persisting through the first phase are responsible for directing the intermediate stage into the second phase of differentiation. Directly comparing the 18 h AHS and the "no HS" samples, two changes stand out: the loss of *bgcn* mRNA and the change in *Rbfox1* splicing to produce the cytoplasmic isoform. Bgcn acts with Bam to antagonise the action of Pumilio (Pum), a translational repressor which blocks differentiation of the GSC (Li et al, 2009; Kim et al, 2010). At 18 h AHS neither Bam nor Bgcn is present, but the alternatively spliced cytoplasmic Rbfox1 isoform has emerged. Rbfox1 is thought to control the transition from the mitotic to post-mitotic stages of differentiation (Tastan et al, 2010) and prevent the reversion into a stem cell fate by repressing translation of *pum* (Carreira-Rosario et al, 2016). We speculate that a second pulse of Bam at 18 h AHS would lead to a very different regulatory trajectory, due to the loss of Bgcn and the presence of cytoplasmic Rbfox1. Furthermore, it is likely that some additional changes at the protein level persist from the first wave to the 18 h inflection point. Quantitative mass spectrometry could be applied to examine these protein level changes during differentiation.

The datasets presented here can be interrogated at a single pathway or gene level to provide insight into a large diversity of biological questions during differentiation and will be a valuable resource for stem cell and developmental biologists. We have uncovered two phases of gene expression change during differentiation, first driving a mitotic phase, then returning to a stem cell-like expression program before promoting terminal differentiation. Many adult stem cells that undergo a transit-amplifying stage to increase cell number before terminal differentiation, including in the larval brain as well as intestine, skin and hematopoietic systems (Homem and Knoblich, 2012; Watt, 2001; Micchelli and Perrimon, 2006; Blanpain and Fuchs, 2006; Comazzetto et al, 2021). Therefore, the gene regulatory mechanisms uncovered here may be broadly applicable to understand fate changes during differentiation.

# Methods

## Resource availability

Further information and requests for resources or reagents should be directed to the lead contact Felipe Karam Teixeira (fk319@cam.ac.uk). Any additional information required to reanalyse the data reported in this paper is available from the lead contact upon request.

## *Drosophila* husbandry and genetics

Unless otherwise stated, stocks and crosses were maintained on standard propionic food at 25 °C for experiments. The *Drosophila melanogaster* stocks used were:

| *Drosophila* stock | genotype | Origin/RRID |
| --- | --- | --- |
| wild type | *w[1118]* | R. Lehmann lab |
| *bam* mutant | *ry[506] e[1] bam[Δ86]/ TM3, ry[RK] Sb[1] Ser[1]* | RRID:BDSC_5427 |
| *bam⁻/⁻,hs-bam* | *w[1118]; nosP-GAL4-NGT40/ CyO; bamΔ86, P{w[+mC] =hs-bam.O}11d/TM6B* | hs-bam construct from D. McKearin |
| Mod FlyFOS | *PBac{fTRG00197.sfGFP-TVPTBF}VK00033* | VDRC v318495 RRID:Flybase_FBst0491406 |
| Dp1 FlyFOS | *PBac{fTRG01340.sfGFP-TVPTBF}VK00033* | VDRC v318850 RRID:Flybase_FBst0492036 |
| Cirl FlyFOS | *PBac{fTRG01155.sfGFP-TVPTBF}VK00033* | VDRC v318773 RRID:Flybase_FBst0491947 |

## Synchronising GSC differentiation

Virgin females (*w[1118]; nosP-GAL4-NGT40/CyO; bamΔ86, P{w[+mC]=hs-bam.O}11d/TM6B*) were crossed to males (*w;; bamΔ86/Tm6C*). Female F1s (*w;nosP-GAL4-NGT40/+;bamΔ86, P{w[+mC]=hs-bam.O}11d/bamΔ86*) were collected overnight and fattened on yeast for three days. Flies were heatshocked in pre-warmed vials without yeast for 1 h at 37 °C, followed by 2 h at 34 °C. Flies were flipped to fresh vials with yeast, returned to 25 °C and time after heat shock (AHS) was measured from this point.

## smFISH probe labelling

smFISH probes (32 probes per gene) were designed using the Stellaris Probe Designer (Biosearch Technologies) and ordered as unlabelled DNA oligos. Labelling was done according to Gaspar et al, 2017 (Gaspar et al, 2017). Briefly, unlabelled ddUTP was conjugated to an ATTO dye NHS ester (ATTO565 or ATTO633, Atto-tec), then the labelled ddUTP was added to the 3'-end of each probe with terminal deoxynucleotidyl transferase. Probes were purified by ethanol precipitation.

## Antibody staining

Ovaries were dissected in cold PBS and fixed for 25 min at room temperature in 4% formaldehyde in 0.3% PBSTX (0.3% Triton-X). Ovaries were washed with 3 × 15 min in 0.3% PBSTX, then incubated in Block (0.2 μg/μl BSA in 0.3% PBSTX) for 1 h at room temperature. Primary antibodies were added to the Block at the appropriate concentration for incubation overnight at 4 °C. The following day, washes and secondary antibody incubation were performed in Block, with the addition of Hoechst 33342 DNA stain (ThermoFisher Scientific) in one wash step.

| Antibody | Concentration | Origin/RRID |
|---|---|---|
| Alpha-spectrin (mouse) | 1:100 | DSHB Cat#3A9(323 or M10-2) RRID:AB_528473 |
| Vasa (rabbit) | 1:1000 | Ruth Lehmann |
| Bam (mouse) | 1:20 | DSHB bamRRID:AB_10570327 |
| Orb (mouse) | 1:200 | DSHB 4H8 RRID:AB_528418 |
| C(3)G (mouse) | 1:500 | 1A8, (Anderson et al, 2005) |
| GFP Booster ATTO 488 | 1:500 | Chromotek (gba488-100) RRID:AB_2631386 |

## smFISH

Ovaries were dissected and fixed as above for IF. After 3× washes in 0.3% PBSTX, samples were transferred to Wash buffer (2× saline sodium citrate (SSC), 10% deionised formamide in nuclease-free water) for 10 min at room temperature. smFISH probes (Table EV1), primary antibodies and phalloidin (Alexa Fluor 405 or 488 Phalloidin, ThermoFisher Scientific) were diluted in Hybridisation buffer (2× SSC, 10% deionised formamide, 20 mM vanadyl ribonucleoside complex, 0.1 mg/ml BSA, competitor (1:50 dilution of 5 mg/ml *E. coli* tRNA and 5 mg/ml salmon sperm ssDNA) in nuclease-free water). Ovaries were incubated in Hybridisation buffer at 37 °C overnight (less than 16 h). Ovaries were washed 3 × 15 min in Wash buffer, then incubated with secondary antibodies in Wash buffer for 2 h at room temperature. Ovaries were finally washed in Wash buffer, with the addition of Hoechst in one wash step.

## Imaging

Samples were mounted in VectaShield mounting media (Vector Laboratories). Images were acquired on a Leica SP8 confocal microscope with a 20× dry objective or 40× oil objective. Image processing was using Fiji (Schindelin et al, 2012).

## Western blot

In total, 30 pairs of ovaries were collected for each sample and lysed in Laemmli buffer with B-mercaptoethanol. Ovaries were homogenised with an electric pestle and then incubated at 95 °C for 5 min. Samples were run on a Novex Value 4–12% Tris-Glycine Mini Protein Gel. Proteins were transferred to a PVDF membrane by wet transfer. Blots were blocked with 5% skimmed milk in TBST (TBS with 1% Tween) for 1 h at room temperature. Primary antibody incubation was at 4 °C overnight, then blots were washed and incubated with secondary antibodies in TBSTS (TBST with 0.01% SDS) for 30 min at room temperature. Blots were imaged with a Licor Odyssey imager.

| Antibody | Concentration | Origin |
|---|---|---|
| Vasa (rabbit) | 1:5000 | Ruth Lehmann |
| Bam (mouse) | 1:500 | DSHB bam RRID:AB_10570327 |

## RNA-seq

Ovaries were dissected for each sample in cold Dissection buffer (PBS 1×, 0.01% Tween 20, 100 µg/ml cycloheximide) and immediately stored at −80 °C after dissection. Frozen samples were homogenised in Polysome extraction buffer (50 mM Tris pH 7.5, 5 mM MgCl$_2$, 150 mM NaCl, 0.5% Triton X-100, 1 mM DTT, 100 µg/ml cycloheximide, protease inhibitor, 25 µ/ml Turbo DNAse) using an electrical pestle and then further disrupted by passing 20 times through a 26-gauge needle. The lysate was centrifuged at 20,000 × *g* for 10 min at 4 °C to pellet tissue debris. Total RNA was isolated by hot phenol–chloroform extraction and quantified using Qubit RNA High Sensitivity Assay Kit (Invitrogen). Overall, 1.5 µg of total RNA was used for rRNA depletion by binding of complementary oligos and treatment with RNase H (Morlan et al, 2012; ElMaghraby et al, 2019). Libraries were generated using NEBNext®Ultra™ Directional RNA Library Prep Kit for Illumina® according to the manufacturer's instructions. Libraries were multiplexed using the NEBNext® Multiplex Oligos for Illumina® and sequenced in paired-end 150-nt long reads on an Illumina NovaSeq 6000.

## Ribo-seq

In total, 250 pairs of ovaries were dissected for each sample in cold Dissection buffer (PBS 1×, 0.01% Tween 20, 100 µg/ml cycloheximide) and stored at −80 °C. As for RNA-seq, frozen samples were homogenised with an electrical pestle in Polysome extraction buffer (50 mM Tris pH 7.5, 5 mM MgCl$_2$, 150 mM NaCl, 0.5% Triton X-100, 1 mM DTT, 100 µg/ml cycloheximide, protease inhibitor, 25 u/ml Turbo DNAse), then passed 20 times through a 26-gauge needle. The lysate was centrifuged at 20,000× *g* for 10 min at 4 °C to pellet tissue debris. 100–140 µg of total RNA was used for Ribo-seq. Total RNA was treated with 1.25 units of RNase I (Ambion) per µg RNA, and the quenched with SuperaseIn (0.8 units per µg RNA). The sample was brought to 1 ml by adding Polysome extraction buffer and subjected to a 3 ml 34% sucrose cushion, by centrifugation at 70,000 rpm for 5.5 h at 4 °C. The resulting pellet was resuspended in nuclease-free water with 1% SDS. RNA was extracted with hot phenol–chloroform and purified by Zymo RNA Clean and Concentrator kit. rRNA was depleted as for the RNA-seq through binding of complementary oligos (ElMaghraby et al, 2019) and treatment with RNase H (Morlan et al, 2012). Size selection was performed on a 15% Urea PAGE Gel, cutting the band corresponding to 28 to 34 bp. RNA was purified by ZYMO small-RNA PAGE recovery kit. RNA was treated with 1 µl of T4 PNK for end repair and libraries were produced with the NEBNext Small RNA Library Kit, according to the manufacturer's instructions. Libraries were size selected at 148–154 nucleotides, corresponding to ligated constructs from 28 to 34 nt RNA fragments. Libraries were multiplexed using the NEBNext® Multiplex Oligos for Illumina® and sequenced in single-end 50-nt long reads on an Illumina HiSeq 4000.

## Data analysis

Trim galore, integrating the trimmer tool cutadapt (Martin, 2011), was used for adapter trimming and quality control of both (paired-

end) RNA-seq and (single-end) Ribo-seq data. Subsequently, random 4 nucleotides (4 N adapters) from both 3′- and 5′-ends, introduced during library preparation, were removed from Ribo-seq reads after excluding redundant reads. Trimmed reads were aligned to non-coding RNA reference sequences (flybase, dmel_r6.39) using Bowtie (Langmead et al, 2009) for Ribo-seq and Bowtie2 (Langmead and Salzberg, 2012) for RNA-seq data and then non-matching reads were mapped to the *Drosophila melanogaster* reference genome dm6 using STAR (Dobin et al, 2013). For quality control and benchmarking of Ribo-seq data we used the R package ribosomeProfilingQC (Ou and Hoye, 2022). Transcript abundance was quantified and differentially expressed genes were identified using Cuffdiff v2.2.1 (Trapnell et al, 2010). Splicing analysis was performed with Jum (version 1.3.12) (Wang and Rio, 2018), after 2-pass mapping with STAR. For analysis of translation efficiency, read counts were generated using featureCounts (Liao et al, 2014). Differences in translation efficiency and classification of gene regulation was analysed using deltaTE (Chothani et al, 2019). All analyses were performed with two samples, each with two biological replicates. For comparison to whole ovary, RNA-seq and Ribo-seq datasets from (Greenblatt and Spradling, 2018) (Data ref: NCBI BioProject PRJNA466150 (2018), three mCherry RNAi control samples) were reanalysed with cufflinks. GO enrichment analysis was performed using FlyMine (Lyne et al, 2007) with Holm–Bonferroni correction. PCA analysis was performed using the prcomp function in R, using scaled variables.

Imaging experiments were repeated for at least three biological replicates. For sequencing experiments, we performed two biological replicates. No statistical methods were used to pre-determine sample size. No statistics were performed on imaging data. Experiments were neither intentionally randomized nor intentionally ordered. Investigators were not blinded to allocation during experiments or outcome assessment.

## Data availability

RNA-seq and Ribo-seq data: Gene Expression Omnibus, GSE246393. Imaging data: BioImage Archive, S-BIAD1009. This paper does not report the original code.

## Peer review information

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

## Acknowledgements

The authors thank Ruth Lehmann and Scott Hawley for reagents and antibodies; Juan Mata and Caia Duncan for technical assistance with Ribo-seq experiments; the Vienna Drosophila Resource Center, and the Bloomington Drosophila Stock Center for fly reagents; AD Hay, KZA Grobicki, PE Andersen, and TH Grønbæk for discussion and comments on the manuscript. The authors also acknowledge the help of the following facilities: the Department of Genetics Fly Facility and the CRUK Cambridge Institute Genomics Core. The authors gratefully acknowledge Dr Ian Clark, Dr Antonina J. Kruppa, Dr Ben Sutcliffe, and Dr Jonathan D. Howe from the Department of Genetics imaging facility for their support and assistance in this work and thank the Wellcome Trust for a strategic award (105602/Z/14/Z). TJS is a Herchel Smith Postdoctoral Fellow, and DG is supported by a Walter Benjamin Postdoctoral Fellowship from Deutsche Forschungsgemeinschaft (GE3407/1-1). FKT is a Wellcome Trust and Royal Society Sir Henry Dale Fellow (206257/Z/17/Z) and is supported by the Human Frontier Science Program (CDA-00032/2018). For the purpose of Open Access, the author has applied a CC BY public copyright licence to any Author Accepted Manuscript (AAM) version arising from this submission.

## Author contributions

**Tamsin J Samuels**: Conceptualization; Data curation; Software; Formal analysis; Funding acquisition; Validation; Investigation; Visualization; Methodology; Writing—original draft; Writing—review and editing. **Jinghua Gui**: Conceptualization; Data curation; Formal analysis; Validation; Investigation; Visualization; Methodology. **Daniel Gebert**: Data curation; Software; Formal analysis; Funding acquisition; Validation; Investigation;

Visualization; Methodology. **Felipe Karam Teixeira**: Conceptualization; Data curation; Formal analysis; Supervision; Funding acquisition; Investigation; Visualization; Methodology; Writing—original draft; Project administration; Writing—review and editing.

## Disclosure and competing interests statement

The authors declare no competing interests.

# Expanded View Figures

**Figure EV1.  RNA-seq in synchronised differentiating GSCs recapitulates changes in gene expression during normal differentiation.**

(**A**) As in Fig. 2A, smFISH in *wild type* germaria (right) was used to validate the RNA-seq results (left). Staining for DNA (Hoechst, blue), actin (phalloidin, gray), fusome (α-spectrin, yellow), and mRNA transcript of interest (smFISH, red and grayscale): (i) *cuff*, (ii) *thymidylate synthase*, (iii) *CG11674*, (iv) *CG32814 (eggplant)*, (v) *CG14545*, (vi) *blanks*. Scale bars are 15 μm. (**B**) Heatmap illustrating RNA-seq expression level across our time course, of marker genes identified by scRNA-seq pseudotime analysis by Rust et al, 2020 (illustrated in Fig. 2m in that paper, and approximate expression domain labelled here left) for each time point from GSC to 16cc. Each row represents one gene and expression level is scaled per gene: black— mean expression across the time course, gold—25% higher expression than the mean, cyan—25% lower expression than the mean. (**C**) The 0.5 h AHS sample was used to exclude 67 genes which showed a significant change of >threefold between the 0.5 AHS and the no HS samples (with expression >10 FPKM in one of these samples). The expression of these 67 genes is shown as log2foldchange at each time point relative to no HS. (**D**) Gene ontology enrichment analysis of the excluded 67 genes shown in (**C**) found a significant enrichment in terms associated with a heat shock response. Colour of the bar indicates *P* value of the enrichment. Background is the whole genome. GO enrichment analysis was performed using FlyMine (Lyne et al, 2007) with Holm–Bonferroni correction.

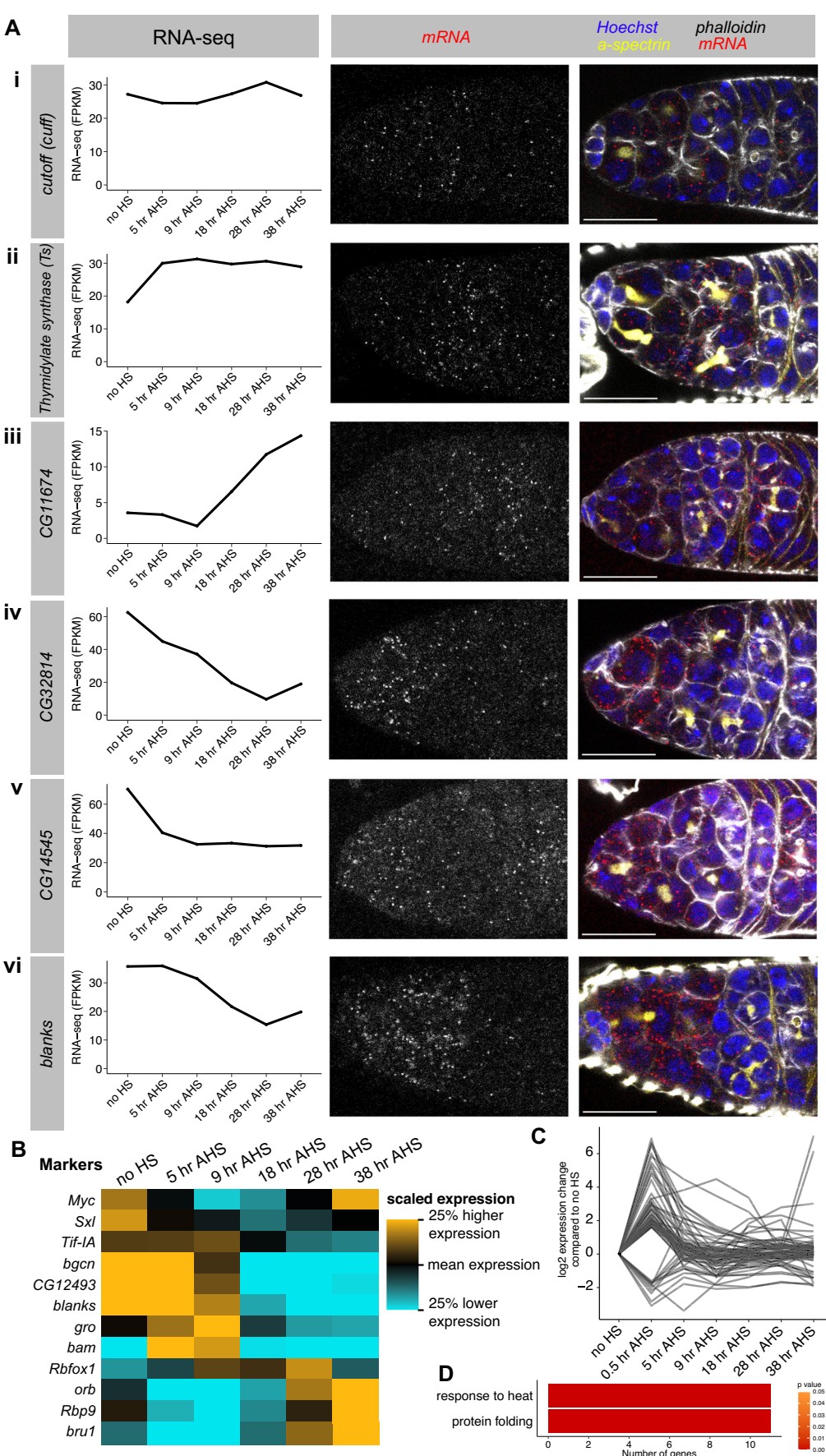

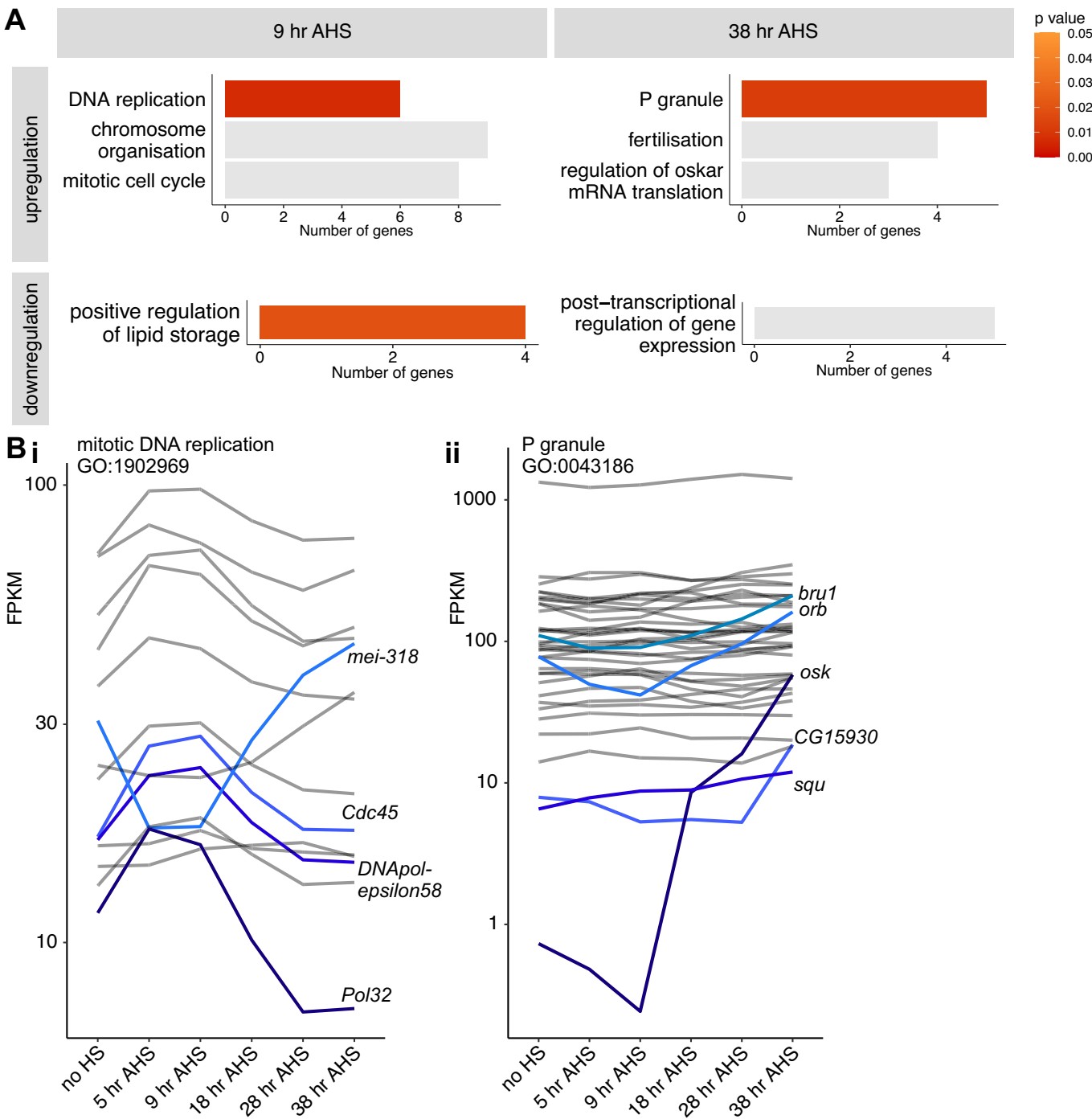

**Figure EV2. RNA-seq reveals changes in mRNA level during differentiation.**

(A) Gene ontology enrichment analysis of genes upregulated or downregulated in the RNA-seq at 9 h AHS or 38 h AHS compared to 'no HS'. Colour of the bar indicates p value of the enrichment, grey = $P > 0.05$. Background is all genes expressed during the time course. GO enrichment analysis was performed using FlyMine (Lyne et al, 2007) with Holm–Bonferroni correction. (B) RNA-seq FPKM for genes in two gene ontology groups: mitotic DNA replication (i) and P granule (ii). Genes with a >1.6-fold change in gene expression are highlighted.

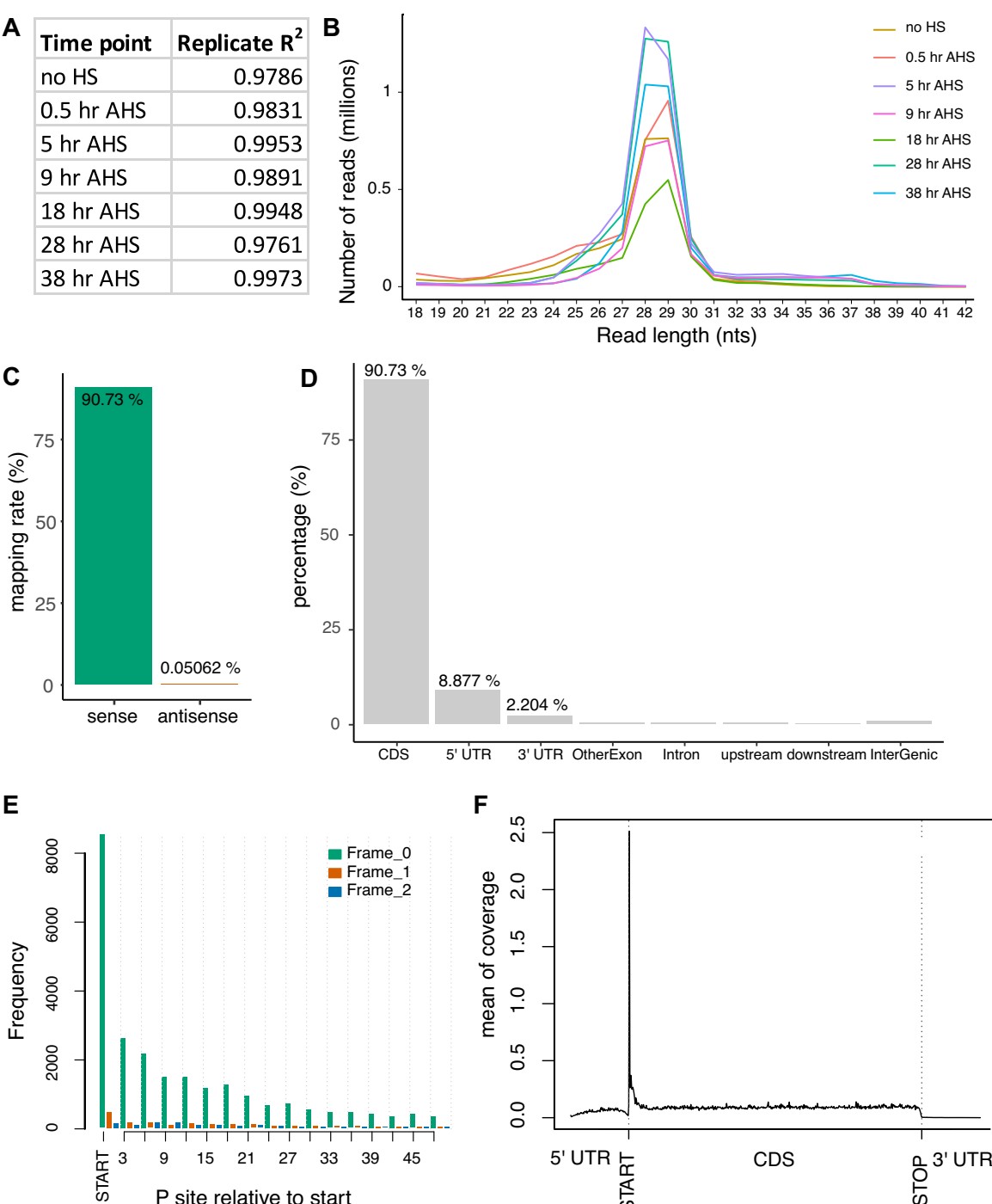

**Figure EV3.  Ribo-seq quality control.**

(A) $R^2$ show good correlation between Ribo-seq replicate samples at each time point. (B) Distribution of read length in nucleotides for Ribo-seq libraries at each time point. (C–F) Ribo-seq quality control for the 'no HS' sample only for illustration. (C) 90.73% of reads mapped to the CDS on the sense strand of genes compared to 0.05% mapping to the CDS on the antisense strand. (D) 8.88% of reads map to the 5' UTR, 2.20% map to the 3' UTR, and mapping to introns and intergenic regions is negligible. (E) P-site mapping shows a strong three nucleotide periodicity, with highest frequency at the start codon. (F) Metagene analysis plot showing read distribution in 5' UTR, CDS and 3' UTR regions, shows consistent coverage across the CDS with the expected bias at the start codon.

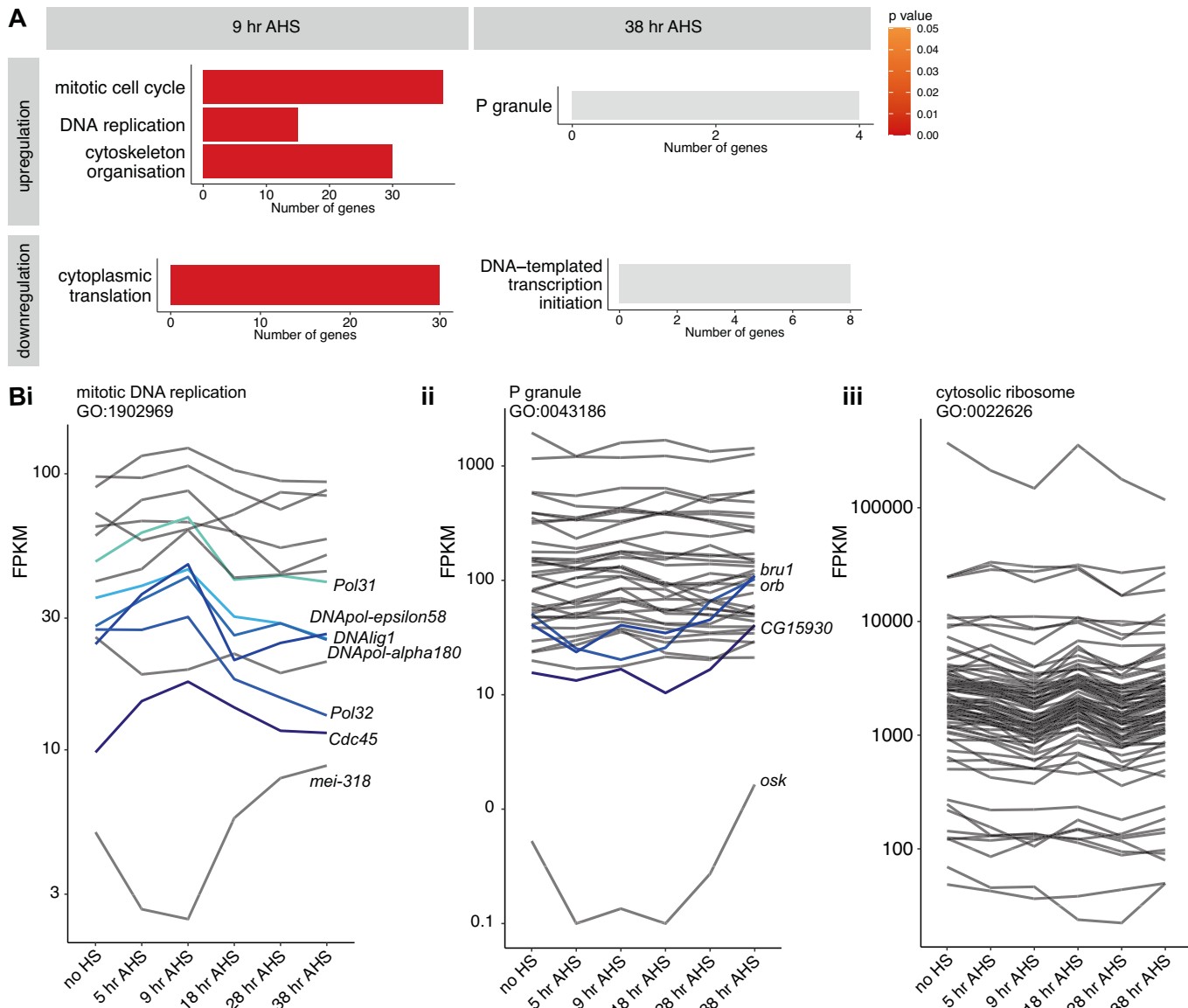

**Figure EV4.  Ribo-seq reveals changes in translation during GSC differentiation.**

(A) Gene ontology enrichment analysis of genes upregulated or downregulated in the Ribo-seq at 9 h AHS or 38 h AHS compared to 'no HS'. Colour of the bar indicates p value of the enrichment, grey = $P > 0.05$. Background is all genes expressed during the time course. GO enrichment analysis was performed using FlyMine (Lyne et al, 2007) with Holm–Bonferroni correction. (B) Ribo-seq FPKM for genes in three gene ontology groups: mitotic DNA replication (i), P granule (ii) and cytosolic ribosome (iii). Genes with a >1.6-fold change in gene expression are highlighted (except in iii due to too many genes).

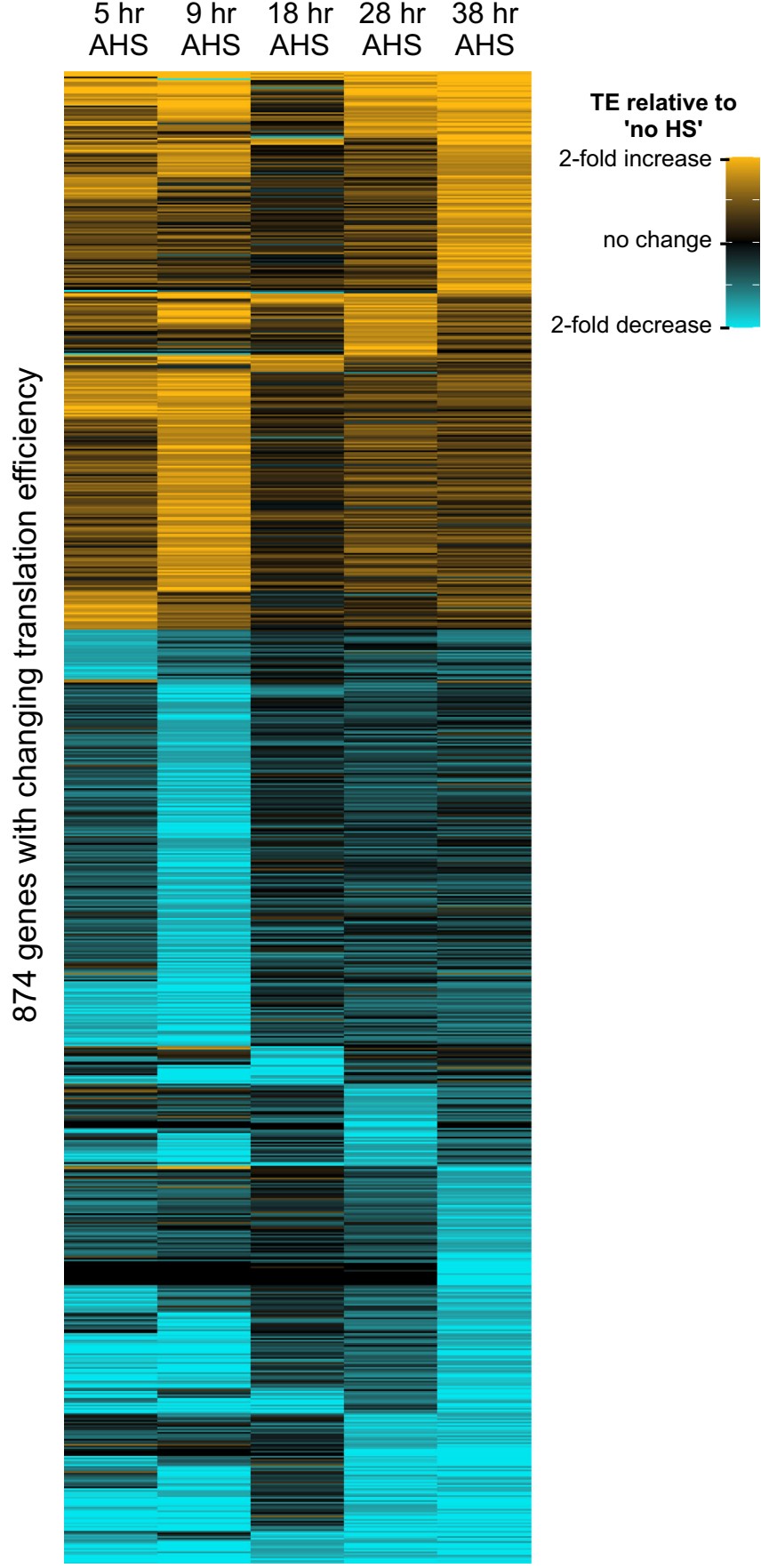

◄ **Figure EV5. Many genes are regulated by TE.**

Heatmap showing fold change in TE, compared to the 'no HS' time point, for the 874 genes which exhibit a significant 1.6-fold change in translation between a given time point and 'no HS'. cyan = twofold decrease compared to 'no HS', black = no change, gold = twofold increase.

