## [Peer Review File · The EMBO Journal]

Two distinct waves of transcriptome and translome changes drive *Drosophila* germline stem cell differentiation

Tamsin Samuels, Jinghua Gui, Daniel Gebert, and Felipe Karam Teixeira

Corresponding author: Felipe Karam Teixeira (fk319@cam.ac.uk)

Review Timeline:

Submission Date:	22nd Nov 23
Editorial Decision:	12th Dec 23
Revision Received:	25th Jan 24
Editorial Decision:	21st Feb 24
Revision Received:	22nd Feb 24
Accepted:	23rd Feb 24

Editor: Ieva Gailite

Transaction Report:

Dear Dr. Karam Teixeira,

Thank you for submitting your manuscript for consideration by the EMBO Journal. We have now received comments from three reviewers, which are included below for your information.

As you will see from the reports, all reviewers find the study of interest, while also pointing out several aspects that would need to be clarified in the final manuscript before they can recommend acceptance of the manuscript. Based on these positive comments, I would like to invite you to address the issues raised by the referees in a revised manuscript.

We generally allow three months as standard revision time. As a matter of policy, competing manuscripts published during this period will not negatively impact on our assessment of the conceptual advance presented by your study. However, please contact me as soon as possible upon publication of any related work to discuss the appropriate course of action. Should you foresee a problem in meeting this three-month deadline, please contact us to arrange an extension.

When preparing your letter of response to the referees' comments, please bear in mind that this will form part of the Review Process File and will therefore be available online to the community. For more details on our Transparent Editorial Process, please visit our website: <https://www.embopress.org/page/journal/14602075/authorguide#transparentprocess>. Please also see the attached instructions for further guidelines on preparation of the revised manuscript.

Please feel free to contact me if you have any further questions regarding the revision. Thank you for the opportunity to consider your work for publication. I look forward to your revision.

With best regards,

Ieva

Ieva Gailite, PhD
Senior Scientific Editor
The EMBO Journal
Meyerohofstrasse 1
D-69117 Heidelberg
Tel: +4962218891309
i.gailite@embojournal.org

We realize that it is difficult to revise to a specific deadline. In the interest of protecting the conceptual advance provided by the work, we recommend a revision within 3 months (11th Mar 2024). Please discuss the revision progress ahead of this time with the editor if you require more time to complete the revisions.

Referee #1:

In this manuscript, the authors report an in-depth analysis of both the transcriptional and translational changes *Drosophila* ovarian germ cells experience as they undergo early differentiation. The authors synchronize germ cell differentiation by inducing a pulse of bam expression in an otherwise bam mutant background. The authors then perform microscopy to determine specific time points after heat shock that correspond to specific developmental stages: no HS (bam mutant), 5 hr (cystoblast, CB), 9 hr (2cc), 18 hr (4cc), 28 hr (8cc) and 38 hr (16cc) AHS. Using these roughly synchronised populations, the authors perform RNA-seq and ribo-seq experiments. Interestingly, RNA-seq analysis shows that in addition to stage specific increases and decreases in mRNA levels, splicing patterns also change during germ cell differentiation. Several examples are given including *Rbfox1* and *HtsRC*, which have important functional consequences. Additional analysis shows that transcriptional start sites also change during the course of early germ cell development. Complementing the RNA-seq with ribo-seq techniques, the authors find that gene expression is dynamically regulated at the level of mRNA translation. Moreover, the regulation of translation accounts for more variability in gene expression across different stages compared to mRNA levels. Based on their data, the authors propose a model whereby germ cells experience two waves of transcriptional and translational remodeling before they form egg chambers.

Overall, the manuscript is well-written, the data are clearly presented, and the findings represent a significant advance. The datasets provided here will be of broad interest and an important resource for the field. In general, I am enthusiastic about the publication of this work. However, several points should be addressed before acceptance of the manuscript for publication.

One key difference between the synchronised differentiation protocol used in this study and what germ cells typically experience is that germ cells in normal ovaries exhibit sustained expression of Bam that persists from the cystoblast to the 8-cell cyst stage. This is an important caveat that should be mentioned/emphasized in the results and discussion. This in no way diminishes the potential impact of the work. However, it would be prudent to rephrase and/or temper some of the conclusions regarding very specific claims regarding "two waves" of regulation, which may arise in part from the synchronization protocol itself. I would also recommend removing "two" from the title.

Along the same lines, several independent groups have shown that developing female (and male) germ cells are capable of dedifferentiating back into GSCs up to and including the 8-cell cyst stage. I recommend mentioning this point in the results and/or discussion when talking about the functional state of germ cells at various stages.

Figure 1E shows an ovariole with sequentially differentiating egg chambers from bam; hs-bam females 5 days AHS. It is surprising that germ cell tumors do not reform in germaria at some frequency after the initial pulse of bam expression. Can the authors comment and/or provide additional images regarding this point?

The rate of cell division in the synchronised populations seems accelerated compared to cells in normal germaria. Can the authors comment on this?

The formation of 16-cell cysts and entrance into meiosis represents a critical and distinct step in differentiation. When do bam; hs-bam germ cells form synaptonemal complexes after bam induction? C(3)G and gamma-H2AV staining of induced germ cell cysts would strengthen the characterization of the synchronization system.

Previous reports found that "In contrast to yeast and mammalian cell lines, *Drosophila* ribosomes are highly sensitive to RNase I, potentially due to their unusual rRNA sequences and structures" (doi.org/10.7554/eLife.01179). However, this study used RNase I and did not detect the same sensitivity. Other groups typically use MNase to avoid this problem, which results in positional uncertainty with P-site mapping and an inability to detect strong 3-nucleotide P-site periodicity. Can the authors provide an explanation for differences between previously published protocols and the one they use here? Any changes to the methodology that now allow for the use of RNase I to characterise *Drosophila* samples should be explained in detail in the material and

methods.

Characterisation of the cytoplasmic Rbfox1 expression pattern during germ cell differentiation was first based on available protein traps and antibodies against the endogenous protein (DOI: 10.1242/dev.050575). Nevertheless, the authors are correct that the splicing patterns were not fully characterized. The authors should consider rephrasing sentences in the corresponding paragraph.

Previous studies have described uORFs and regulated stop codon read-through in *Drosophila*. Do the authors observe changes in these features during germ cell differentiation? If so, the authors should consider describing these changes in the text and presenting the data in a figure or supplemental figure.

Page 8 typo- Begonial cell neoplasm (bgcn) should be benign gonial cell neoplasm.

Referee #2:

This manuscript utilized a simple but efficient method to synchronize the differentiation of *Drosophila* female germline stem cells (GSCs) *in vivo*. It performed in-depth transcriptome and translome analyses to investigate the differentiation process of GSCs over time. The translome analysis revealed widespread and dynamic changes in mRNA levels, promoter usage, exon inclusion, and translation efficiency during GSC differentiation into 16-cell cysts. These findings demonstrate that the *in vivo* differentiation program relies on distinctly regulated and developmentally sequential waves, rather than a unidirectional accumulation of changes. This provides valuable insights into the important role of translational regulation during development. The datasets generated in this manuscript can be interrogated at the single pathway or gene level, which may provide insights into a wide range of biological questions related to GSC differentiation. As a result, these findings should be of significant interest to researchers studying stem cell biology. However, there are some concerns that may affect the validity of the authors' conclusions. These concerns should be addressed before publication.

Specific points

1. According to the results, there are two distinct waves of gene expression changes observed at 5-9 hr and 28-38 hr after heat shock (AHS). This suggests that the process of differentiation into a new fate involves two separate phases. It is possible that early differentiation and terminal differentiation are regulated by different transcription factors. There are several specifically upregulated genes identified in both the 5-9 hr and 28-38 hr AHS time points (Figure 2C-2E). Are there any key transcription factors that play an important role in early or terminal differentiation?
2. When a daughter cell is excluded from the germline stem cell niche, bam is transcriptionally upregulated, initiating the process of differentiation. The expression of the bam protein was notably increased at the 0.5 hr after heat shock (AHS) time point (Figure 1C). Therefore, it is possible that the 0.5 hr AHS time point represents the cystoblast (CB) stage, rather than the 5 hr AHS time point.
3. The significant dots on the volcano plot are all almost symmetrically distributed (Figure 3A), what does this symmetry mean?
4. A large group of differentially expressed genes was detected at a threshold of 10 FPKM (Figure 2B, 4A). Why was a 1.6-fold variation selected for differential expression analysis? Either a 1.6-fold change or 10 FPKM is very low and uncommonly used. If the threshold is raised, will the transcriptome and the translome show the same change?
5. Biological replicates are important for identifying differentially expressed genes with confidence, and three biological replicates are a common practice. However, this study seems to have conducted most of their experiments with only two biological replicates.
6. It is necessary to perform protein staining rather than mRNA detection to identify late translation shown in Figure 6C.

Referee #3:

Samuels/Gui et al review

In this manuscript, Samuels/Gui et al develop a method to synchronize oogenic germ cells by first preventing differentiation through a knockout of the differentiation factor bam and then subsequently transiently expressing bam through a heatshock transgene. This system, well-validated by the authors, enables bulk amounts of stage-specific germ cells to be collected for RNA-seq and Ribo-seq. The authors subsequently use these datasets to identify surprising waves of transcription and translation, as well as differentiation-stage-specific isoforms. Together, this allows for generation of a model in which the stem cell transcriptional/translatable landscape changes considerably after one mitosis, but then returns to a stem cell like state in two more division steps before proceeding with terminal differentiation.

This paper was a genuine joy to read. Too frequently, papers presenting large sequencing datasets fail to *in vivo* validate their findings and fall prey to artifactual conclusions. Here, the authors carefully and consistently test their sequencing findings *in vivo* and do not make any functional conclusions beyond what their data suggests. Instead, through careful work, the authors are

able to strongly support their conclusion that the differentiation program relies on sequential waves of transcription and translation.

The paper is of high enough quality to be published as-is. Below are some optional suggestions that may improve the paper (but are not required for publication).

Experiments/new figures:

- The smFISH/reporter validation is fantastic. It would be even better if it were expanded somewhat. Some examples of places it could be expanded:
 - o An important note the authors make is that there is not a unique, stem cell specific transcriptional program, and rather, the GSC program is shared with other cell types. They do, however, suggest that 3 genes (CG17127, CG14545, and CG11892) might serve as putative positive or negative markers of GSCs. FISH on one or all of these would be very exciting.
 - o Likewise, the splice-specific data is very interesting. Isoform-specific FISH, if possible, on WT ovaries demonstrating that alternative splice isoforms are present in different stages would be wonderful.
 - o The simultaneous examination of RNA and protein levels with FlyFOS transgenes is great. It would be even better if they showed an example in which the transcript wave precedes the translational wave, or really another expression dynamics that aren't flat.
- The authors say that the majority of their Ribo-seq reads have the expected ribosome footprint, but don't show a histogram of the footprint size. This would be nice to see in a supplemental fig.
- The PCA of the sequencing libraries is a beautifully visually striking piece of data, and it's so clear that noHS is most similar to 18hr HS on PC1. It would be great if the authors performed an analysis of what genes comprise PC1 vs. PC2 (in the positive and negative direction), and if the genes comprising PC1 make sense as re-acquired or re-suppressed stem factors at 18 hours.

Text/clarity changes

- In general, the microscopy images could be bigger, especially 1D. Single-channel images for 1E (bottom) would be great. It's unclear from the image if orb is restricted to a single cell in the equivalent cell stage in the hs-bam vs. the wild type in the image as shown.
- The comparison of their data to existing trajectory inference data from single cell sequencing data is a wonderful control to determine how well their synchronization system recapitulates normal development (S1B). It would be a great aid to the reader if the 12 genes selected were labeled as to in what stage of differentiation they are expressed (in the scRNA-seq data).
- Regressing out genes whose expression changes due to heatshock is another important control the authors perform. For clarity, the authors should state that they generated RNA-seq libraries from wild-type ovaries at 0.5 hr AHS.
- The authors should clarify in their text or methods what background was set for the GO term analysis (e.g., was the background from pooled RNAseq all the HS conditions, or from WT ovary transcriptomes?)

Response to Referees

We thank the reviewers for their positive feedback and helpful suggestions. As you will see below, we have addressed all of the comments - primarily with the suggested text alterations/clarifications as well as adding a staining of C(3)G to define the onset of meiosis better. All the points raised have been addressed either by changes in the text/figures and/or by the discussions below. We believe that the changes we have made have improved our paper and have addressed all comments.

Referee #1:

In this manuscript, the authors report an in-depth analysis of both the transcriptional and translational changes *Drosophila* ovarian germ cells experience as they undergo early differentiation. The authors synchronize germ cell differentiation by inducing a pulse of bam expression in an otherwise bam mutant background. The authors then perform microscopy to determine specific time points after heat shock that correspond to specific developmental stages: no HS (bam mutant), 5 hr (cystoblast, CB), 9 hr (2cc), 18 hr (4cc), 28 hr (8cc) and 38 hr (16cc) AHS. Using these roughly synchronised populations, the authors perform RNA-seq and ribo-seq experiments. Interestingly, RNA-seq analysis shows that in addition to stage specific increases and decreases in mRNA levels, splicing patterns also change during germ cell differentiation. Several examples are given including *Rbfox1* and *HtsRC*, which have important functional consequences. Additional analysis shows that transcriptional start sites also change during the course of early germ cell development. Complementing the RNA-seq with ribo-seq techniques, the authors find that gene expression is dynamically regulated at the level of mRNA translation. Moreover, the regulation of translation accounts for more variability in gene expression across different stages compared to mRNA levels. Based on their data, the authors propose a model whereby germ cells experience two waves of transcriptional and translational remodeling before they form egg chambers.

Overall, the manuscript is well-written, the data are clearly presented, and the findings represent a significant advance. The datasets provided here will be of broad interest and an important resource for the field. In general, I am enthusiastic about the publication of this work. However, several points should be addressed before acceptance of the manuscript for publication.

One key difference between the synchronised differentiation protocol used in this study and what germ cells typically experience is that germ cells in normal ovaries exhibit sustained

expression of Bam that persists from the cystoblast to the 8-cell cyst stage. This is an important caveat that should be mentioned/emphasized in the results and discussion. This in no way diminishes the potential impact of the work. However, it would be prudent to rephrase and/or temper some of the conclusions regarding very specific claims regarding "two waves" of regulation, which may arise in part from the synchronization protocol itself. I would also recommend removing "two" from the title.

Response: This is an astute point - the single burst of Bam induced by our protocol does not persist as long as in *wild type* differentiation. We observed background levels of Bam by 18 hr AHS (4cc), while *wild type* Bam persists to the 8cc. Bam is the driver of the first wave and therefore the shorter period of expression may lead to a shortened timing of the first wave. However, our validations in the *wild type* suggest that the timing differences in Bam expression do not lead to substantially different expression patterns from what was observed in the RNA-seq analysis. We have adapted the text to point directly to this difference in Bam expression timing both in the results (relating to Figure 1C and 1D, p7) and Discussion sections (first paragraph, p24).

Along the same lines, several independent groups have shown that developing female (and male) germ cells are capable of dedifferentiating back into GSCs up to and including the 8-cell cyst stage. I recommend mentioning this point in the results and/or discussion when talking about the functional state of germ cells at various stages.

Response: This is a very interesting point in connection with our model and we have added this to the discussion (first paragraph, p24).

Figure 1E shows an ovariole with sequentially differentiating egg chambers from bam; hs-bam females 5 days AHS. It is surprising that germ cell tumors do not reform in germaria at some frequency after the initial pulse of bam expression. Can the authors comment and/or provide additional images regarding this point?

Response: At 5 days AHS the ovary is a mixture of different stages of development - indeed we do see reforming of tumours in some germaria (see rebuttal Figure 1).

Response Figure 1: Example *bam^{-/-}, hs-bam* ovaries at 5 days after heat shock (AHS). Stained with DAPI (blue, DNA) and alpha-spectrin (green, fusome). On the right, a more magnified image of the anterior tip of ovaries shows the reformation of germ cell tumours.

The rate of cell division in the synchronised populations seems accelerated compared to cells in normal germaria. Can the authors comment on this?

Response: Recently published quantitations of cell cycle length in developing cysts do not differ greatly from our observations (approximately: GSC = 20 hr, CB = 12 hrs, 2cc = 10 hrs, 4cc = 7 hrs, 8cc = 6 hrs, Rubin et al., 2022, doi.org/10.1073/pnas.2207660119). We observe enrichment of 2cc at 9 hrs after the heat shock induces Bam expression, and 4cc a further 9 hrs later. Importantly, the cells are not synchronised for their phase of the cell cycle prior to the induction of Bam and therefore we expect continued heterogeneous timing of divisions. The first wave of genes upregulated downstream of Bam induction is enriched for cell cycle genes, and therefore it is possible that differing dosage of Bam in our system compared to *wild type* may drive some differences in cell cycle in the early stages of differentiation.

The formation of 16-cell cysts and entrance into meiosis represents a critical and distinct step in differentiation. When do *bam*; *hs-bam* germ cells form synaptonemal complexes after *bam* induction? C(3)G and gamma-H2AV staining of induced germ cell cysts would strengthen the characterization of the synchronization system.

Response: This is a useful suggestion: we have now performed C(3)G staining on samples at 38 hr AHS (our latest time point for the RNA-seq and Ribo-seq experiments) and additionally at 48 hr AHS (**Figure 1F**). We found that C(3)G is lowly expressed at 38 hr AHS, but does not form synaptonemal complex until 48 hr AHS. Therefore, our time course up to 38 hr AHS encompasses the span of GSC differentiation prior to meiosis.

Previous reports found that "In contrast to yeast and mammalian cell lines, *Drosophila* ribosomes are highly sensitive to RNase I, potentially due to their unusual rRNA sequences and structures" (doi.org/10.7554/eLife.01179). However, this study used RNase I and did not detect the same sensitivity. Other groups typically use MNase to avoid this problem, which results in positional uncertainty with P-site mapping and an inability to detect strong 3-nucleotide P-site periodicity. Can the authors provide an explanation for differences between previously published protocols and the one they use here? Any changes to the methodology that now allow for the use of RNase I to characterise *Drosophila* samples should be explained in detail in the material and methods.

Response: During the early development of Ribo-seq protocols in *Drosophila* there were issues with ribosome sensitivity to RNase I (Dunn et al., 2013). MNase has been used as an alternative but has the disadvantages in detection mentioned by the reviewer. More recently, there has been further analysis of such issues and the establishment of extensively validated protocols using RNase I - notably from the Aspden Group (Aspden et al., 2014, Patraquim et al., 2020).

Characterisation of the cytoplasmic Rbfox1 expression pattern during germ cell differentiation was first based on available protein traps and antibodies against the endogenous protein (DOI: 10.1242/dev.050575). Nevertheless, the authors are correct that the splicing patterns were not fully characterized. The authors should consider rephrasing sentences in the corresponding paragraph.

Response: We have updated the results text to better reflect the characterisation of Rbfox1 expression done by Tastan et al. and added the citation to that section (p12 paragraph 2).

Previous studies have described uORFs and regulated stop codon read-through in *Drosophila*. Do the authors observe changes in these features during germ cell differentiation? If so, the authors should consider describing these changes in the text and presenting the data in a figure or supplemental figure.

Response: This is an interesting suggestion, which we hadn't previously examined. To explore this possibility, we looked for Ribo-seq reads in 5' UTRs containing an ATG start codon upstream of the main CDS. We found ~4000 genes in this category (note, a previous study identified over 32,000 individual uORFs with some ribosome occupancy during *Drosophila* development (Zhang et al., 2018)). To narrow down this list, we focussed on those with a significant change during our time course and identified that 388 showed a significant change in Ribo-seq reads between any two samples in our time course. However, scanning these manually, we did not identify any patterns of changes in key genes or

correlating with changes in the translation efficiency of the main ORF. However, this was a very preliminary examination, and we hope others with an interest in this question may be able to explore further, as the data is publicly available.

We also manually examined the 283 genes with predicted stop codon read-through (Jungreis et al., 2011) and found that 114 of these genes are translated in our dataset. However, we only found evidence of read-through in two cases (*hdc* and *ctp*), which do not seem to change during differentiation.

Page 8 typo- Begonial cell neoplasm (bgcn) should be benign gonial cell neoplasm.

Response: We have corrected this typo.

Referee #2:

This manuscript utilized a simple but efficient method to synchronize the differentiation of *Drosophila* female germline stem cells (GSCs) in vivo. It performed in-depth transcriptome and translome analyses to investigate the differentiation process of GSCs over time. The translome analysis revealed widespread and dynamic changes in mRNA levels, promoter usage, exon inclusion, and translation efficiency during GSC differentiation into 16-cell cysts. These findings demonstrate that the in vivo differentiation program relies on distinctly regulated and developmentally sequential waves, rather than a unidirectional accumulation of changes. This provides valuable insights into the important role of translational regulation during development.

The datasets generated in this manuscript can be interrogated at the single pathway or gene level, which may provide insights into a wide range of biological questions related to GSC differentiation. As a result, these findings should be of significant interest to researchers studying stem cell biology. However, there are some concerns that may affect the validity of the authors' conclusions. These concerns should be addressed before publication.

Specific points

1. According to the results, there are two distinct waves of gene expression changes observed at 5-9 hr and 28-38 hr after heat shock (AHS). This suggests that the process of differentiation into a new fate involves two separate phases. It is possible that early differentiation and terminal differentiation are regulated by different transcription factors. There are several specifically upregulated genes identified in both the 5-9 hr and 28-38 hr

AHS time points (Figure 2C-2E). Are there any key transcription factors that play an important role in early or terminal differentiation?

Response: We are also very interested in exploring the regulators driving each wave of expression changes, which may include both transcription factors and RBPs. We have described the suggested role of various RBPs that change in level during differentiation. While some annotated transcription factors (or annotated DNA binding proteins) are included in each wave of changes, none stood out as obvious candidate regulators and further work will be required to identify these regulatory networks.

2. When a daughter cell is excluded from the germline stem cell niche, bam is transcriptionally upregulated, initiating the process of differentiation. The expression of the bam protein was notably increased at the 0.5 hr after heat shock (AHS) time point (Figure 1C). Therefore, it is possible that the 0.5 hr AHS time point represents the cystoblast (CB) stage, rather than the 5 hr AHS time point.

Response: It is correct to observe that Bam is already very highly upregulated at 0.5 hr AHS and any immediate effects of Bam might already begin to be observed at this early time point. Indeed each stage of differentiation (CB, 2cc, 4cc etc.) will span a period of several hours, but for experimental practicality, we had to choose a single time point for each. Our imaging analysis suggests that the 5 hr AHS samples primarily represent individual cells (CBs). Therefore, we opted to use the 0.5 hr AHS time point to identify and exclude the rapid and dramatic changes downstream of the heat shock. It is important to note that these changes have substantially resolved by 5 hr AHS (Figure EV1C).

3. The significant dots on the volcano plot are all almost symmetrically distributed (Figure 3A), what does this symmetry mean?

Response: In Figure 3A, each dot represents the change in usage of a single splice site. When a given splice site becomes relatively more used, it is generally the case that another alternative splice site becomes relatively less used (though in more complex gene structures, the balance between several sites may be concomitantly shifted). Therefore, a splice site identified on one side of the graph is most frequently mirrored by an alternative splice site on the other side. We have added a sentence to the text to clarify this (paragraph 1, p12).

4. A large group of differentially expressed genes was detected at a threshold of 10 FPKM (Figure 2B, 4A). Why was a 1.6-fold variation selected for differential expression analysis? Either a 1.6-fold change or 10 FPKM is very low and uncommonly used. If the threshold is raised, will the transcriptome and the translome show the same change?

Response: Our expression thresholds were chosen based on our validations back to the *wild type* differentiation using smFISH. As discussed in the manuscript (final paragraph p8), for genes expressed at ~10 FPKM, we consistently observe low numbers of transcripts in *wild type*, so we considered these as 'expressed'. However, for genes with <10 FPKM signal, we only sporadically detected meaningful signals by smFISH. We also validated many genes with changes over differentiation, and observed that fold-changes extracted from seq data appeared to be dampened when compared to what was observed in *wild type* by smFISH (see Figure EV1A). Therefore, we opted for a relatively low threshold of 1.6-fold to maximise the input for downstream analysis. In addition to validating such changes by smFISH, it is important to note that modifying the thresholds of expression level or fold change alters the number of genes included in different groupings, but does not change the overall interpretation of the data.

5. Biological replicates are important for identifying differentially expressed genes with confidence, and three biological replicates are a common practice. However, this study seems to have conducted most of their experiments with only two biological replicates.

Response: Three biological replicates would be ideal but we were limited by multiple factors: 1) acquiring sufficient material required a lot of dissection, 2) with different time points resulting in night time heat shocks, dissections and sample processing, and 3) the cost of the experiment. For this reason, we performed thorough validations (as recognised by referee #3), comparing changes to *wild type* differentiation.

6. It is necessary to perform protein staining rather than mRNA detection to identify late translation shown in Figure 6C.

Response: The goal of Figure 6 is to show how some genes are expressed at the mRNA level, but are translationally repressed at early stages of differentiation, before being translated later. In Figure 6C, the protein stainings for the genes showing late translation have been well characterised by others, so we focused on performing the smFISH experiments to demonstrate the presence of mRNA transcripts at each stage. Prompted by the referee' comment, we have now added the reference for antibody staining of Mtrm (Xiang et al., 2007) (p18, final paragraph)- we had already cited work with protein stainings for Mei-218 (Manheim et al., 2002), Smaug (Tadros et al., 2007) in the original version of the manuscript.

Referee #3:

Samuels/Gui et al review

In this manuscript, Samuels/Gui et al develop a method to synchronize oogenic germ cells by first preventing differentiation through a knockout of the differentiation factor bam and then subsequently transiently expressing bam through a heatshock transgene. This system, well-validated by the authors, enables bulk amounts of stage-specific germ cells to be collected for RNA-seq and Ribo-seq. The authors subsequently use these datasets to identify surprising waves of transcription and translation, as well as differentiation-stage-specific isoforms. Together, this allows for generation of a model in which the stem cell transcriptional/translatable landscape changes considerably after one mitosis, but then returns to a stem cell like state in two more division steps before proceeding with terminal differentiation.

This paper was a genuine joy to read. Too frequently, papers presenting large sequencing datasets fail to in vivo validate their findings and fall prey to artifactual conclusions. Here, the authors carefully and consistently test their sequencing findings in vivo and do not make any functional conclusions beyond what their data suggests. Instead, through careful work, the authors are able to strongly support their conclusion that the differentiation program relies on sequential waves of transcription and translation.

The paper is of high enough quality to be published as-is. Below are some optional suggestions that may improve the paper (but are not required for publication).

Experiments/new figures:

- The smFISH/reporter validation is fantastic. It would be even better if it were expanded somewhat. Some examples of places it could be expanded:
 - o An important note the authors make is that there is not a unique, stem cell specific transcriptional program, and rather, the GSC program is shared with other cell types. They do, however, suggest that 3 genes (CG17127, CG14545, and CG11892) might serve as putative positive or negative markers of GSCs. FISH on one or all of these would be very exciting.

Response: We have shown smFISH against *CG14545* in Figure EV1Av, where it appears to be enriched in the stem cell but is also expressed in the early differentiating cysts. This is supported by the RNA-seq result which shows a decrease in expression from 5hr AHS, but expression remains above 30 FPKM throughout the time course. We have now referred back to this image when discussing these three gene (first paragraph p11). *CG17127* produces a short transcript (~450 bases), which is too small to design smFISH probes. We

had previously designed probes for CG11892, but these resulted in a very high background and it was not possible to identify single transcripts.

o Likewise, the splice-specific data is very interesting. Isoform-specific FISH, if possible, on WT ovaries demonstrating that alternative splice isoforms are present in different stages would be wonderful.

Response: This is an excellent idea, but it is unfortunately not possible with our smFISH method due to the length of sequence required to make probes (as above). In future, HCR may be an alternative method to explore the isoform differences *in vivo*.

o The simultaneous examination of RNA and protein levels with FlyFOS transgenes is great. It would be even better if they showed an example in which the transcript wave precedes the translational wave, or really another expression dynamics that aren't flat.

Response: This is a great suggestion and indeed we previously explored doing this experiment. However, the collection of FlyFOS lines is somewhat limited, is not enriched for 'germline genes' and we avoided genes which have been previously studied and published. Therefore, we weren't able to find additional good candidate lines for the experiment.

- The authors say that the majority of their Ribo-seq reads have the expected ribosome footprint, but don't show a histogram of the footprint size. This would be nice to see in a supplemental fig.

Response: The histogram is shown in Figure EV3B for each time point (drawn as lines to allow each time point to be shown on one graph).

- The PCA of the sequencing libraries is a beautifully visually striking piece of data, and it's so clear that noHS is most similar to 18hr HS on PC1. It would be great if the authors performed an analysis of what genes comprise PC1 vs. PC2 (in the positive and negative direction), and if the genes comprising PC1 make sense as re-acquired or re-suppressed stem factors at 18 hours.

Response: As suggested, we have previously explored the genes contributing to PC1 and PC2 in both the RNA-seq and Ribo-seq. While some of these genes do make logical sense together with our model (e.g. CycB), the majority do not have such a clear explanation, so we don't think picking out these gene names adds enough to the paper to integrate it into the text.

Text/clarity changes

- In general, the microscopy images could be bigger, especially 1D. Single-channel images

for 1E (bottom) would be great. It's unclear from the image if orb is restricted to a single cell in the equivalent cell stage in the *hs-bam* vs. the wild type in the image as shown.

Response: We have made the suggested changes to Figure 1 - increasing imaging size and showing single channel of Orb.

- The comparison of their data to existing trajectory inference data from single cell sequencing data is a wonderful control to determine how well their synchronization system recapitulates normal development (S1B). It would be a great aid to the reader if the 12 genes selected were labeled as to in what stage of differentiation they are expressed (in the scRNA-seq data).

Response: While it is difficult to pinpoint the precise expression boundaries of the marker genes from pseudotime analysis, we have added approximate expression domain labels in Figure EV1B, to assist the reader.

- Regressing out genes whose expression changes due to heatshock is another important control the authors perform. For clarity, the authors should state that they generated RNA-seq libraries from wild-type ovaries at 0.5 hr AHS.

Response: We agree our writing here was not clear, the 0.5 hr AHS sample was generated using *bam*^{-/-}, *hs-bam* flies to identify the immediate changes induced by the heat shock. We have modified the text to clarify this (first paragraph, p10).

- The authors should clarify in their text or methods what background was set for the GO term analysis (e.g., was the background from pooled RNAseq all the HS conditions, or from WT ovary transcriptomes?)

Response: We have added information about the GO enrichment analysis to the methods and the figure legends. The background for the heat shock genes (Figure EV1D) was the whole genome, while the background for the genes changing at different time points (Figure EV2A and EV4A) was from genes expressed at any point in our time course.

Dear Felipe,

Thank you for submitting a revised version of your manuscript. Your study has now been seen by all original referees, who find that their previous concerns have been addressed and now recommend acceptance of the manuscript. There now remain only a few editorial points that need addressing before I can extend formal acceptance of the manuscript:

1. Please submit up to five keywords.
2. Email to Jinghua Gui (jg889@cam.ac.uk) did not reach the addressee, please check and correct.
3. Please make sure that the order of the sections in the manuscript is as follows: abstract, introduction, results, discussion, materials & methods, data availability section, acknowledgments, disclosure statement and competing interests, references, main figure legends, tables, expanded figure legends.
4. CRediT has replaced the traditional author contributions section because it offers a systematic, machine-readable author contributions format that allows for more effective research assessment. Please remove the Authors Contributions from the manuscript and use the free text boxes beneath each contributing author's name in our online submission system to add specific details on the author's contribution. More information is available in our guide to authors.
5. Please rename "Conflict of interest" section into "Disclosure and competing interests statement" (further info: <https://www.embopress.org/page/journal/14602075/authorguide#conflictsofinterest>).
6. Please move the Data Availability section to the end of Materials and Methods. Please add please add a resolvable link to the GSE246393 dataset. More information about the format of this section can be found here: <https://www.embopress.org/page/journal/14602075/authorguide#dataavailability>.
7. Tables EV1-8 should be renamed Dataset EV1-8 and the files should be uploaded as datasets.
8. Please rename Reagent Table into Table EV1 and add a legend to the file, e.g., in a separate tab.
9. Our data editors have flagged the following issues in figure legends that need correcting:
 - Please indicate the statistical test used for data analysis in the legends of figures 3a (i-ii); EV1d; EV2a; EV4a.
 - Please note that for heatmap present in figures 2c; 5a; EV1b; EV5, a numbered scale bar is not provided. This needs to be rectified.

With best wishes,

Ieva

We realize that it is difficult to revise to a specific deadline. In the interest of protecting the conceptual advance provided by the work, we recommend a revision within 3 months (21st May 2024). Please discuss the revision progress ahead of this time with the editor if you require more time to complete the revisions.

Referee #1:

The authors have addressed my comments and included several clarifications that strengthen the original manuscript. This interesting study expands our knowledge of how mRNA translation programs change during Drosophila germ cell differentiation.

The manuscript is ready for publication.

Referee #2:

The authors have adequately addressed the previous comments, and I don't have any further comments.

Referee #3:

The authors have done wonderful work, worthy of publication! I have no additional comments or requests for review.

The authors addressed the remaining editorial issues.

Dear Felipe,

Thank you for addressing the final editorial issues. I am now pleased to inform you that your manuscript has been accepted for publication.

I will look into the synopsis text in the next couple of days and let you know if any edits to the journal style are needed.

If you have any questions, please do not hesitate to contact the Editorial Office. Thank you for this contribution to The EMBO Journal and congratulations on a great paper!

Best wishes,

Ieva
